# Efficient Multi-Step Reinforcement Learning with Expectation-Maximization Bootstrapping

## Abstract

Multi-step reinforcement learning (RL) improves agent performance by propagating temporal information across long time lags between actions and consequences through bootstrapping. The key challenge is how to aggregate information from different bootstrapping steps to enable fast learning while maintaining stability. Many existing multi-step RL methods (e.g., Retrace($\lambda$)) primarily focus on the bias–variance tradeoffs but do not explicitly select bootstrapping steps to balance salience and stability (S&S). We first analyze S&S in multi-step RL, and introduce a novel corresponding metric to quantify different bootstrapping steps. Viewing bootstrapping steps as the latent variables, our Expectation-Maximization (EM) Bootstrapping (EMB) formulates multi-step RL as an EM procedure, alternating between the E-step: estimating expectations under predefined posterior weights to measure the S&S of bootstrapping steps, and the M-step: using these estimated expectations to guide the selection of bootstrapping steps. This yields a new return-based Bellman operator EMB($\lambda$). We theoretically establish its convergence and optimality properties. Empirical results on the Atari Learning Environment demonstrate that EMB($\lambda$) significantly outperforms existing multi-step RL methods in both learning efficiency and final performance, matching the performance of Retrace($\lambda$) with approximately $50\%$ fewer samples on the Atari-10 suite.

## 1 Introduction

Reinforcement learning (RL) (Kaelbling et al., 1996; Sutton & Barto, 2018) has proved its mettle in solving sequential decision-making problems such as board games (Tesauro, 1994; Silver et al., 2016), video games (Mnih et al., 2015; Berner et al., 2019), and complicated control tasks (Akkaya et al., 2019). Traditional RL uses temporal difference (TD) learning to update the value estimation by bootstrapping from previously learned predictions, a learning process rooted in the Bellman operator (Sutton & Barto, 2018). Specifically, the selection of the TD target strongly influences the performance, efficiency, and stability of the RL methods. For instance, one-step TD learning is stable but inefficient, as it restricts the propagation of the critical information across the long-term temporal distances (Wang et al., 2024b). While using the Monte Carlo (MC) target enables faster temporal information propagation, resulting in inefficient and unstable learning due to high variance and the latency of waiting for episode termination. Therefore, determining appropriate TD targets has become a central challenge in the development of stable and efficient RL algorithms.

Beyond the one-step TD target, multi-step methods aim to accelerate learning by balancing the stability of one-step updates with the faster propagation of Monte Carlo returns, including $n$-step methods (Sutton & Barto, 2018; Mnih et al., 2016; Hessel et al., 2018) and return-based methods (Sutton, 1988; Harutyunyan et al., 2016; Munos et al., 2016). Specifically, Retrace($\lambda$) (Munos et al., 2016) has been the most successful one, truncating importance sampling ratios of the trace, ensuring safe and efficient learning from the off-policy data. Although some works (Espeholt et al., 2018; Badia et al., 2020a) aim to further improve Retrace($\lambda$), they fail to revolutionize its fundamental principle, and its algorithmic properties remain largely unchanged.

However, existing multi-step RL methods primarily focus on the bias-variance tradeoff, while return-based methods (e.g., Retrace($\lambda$)) struggle to gain performance improvement as the maximum bootstrapping steps $N$ increases, and $n$-step methods (e.g., $n$-step DQN) remain highly sensitive to the choice of bootstrapping step $n$, as demonstrated in the motivation experiments (Figure 1). Specifically,

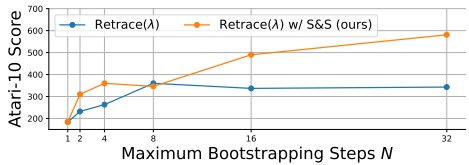
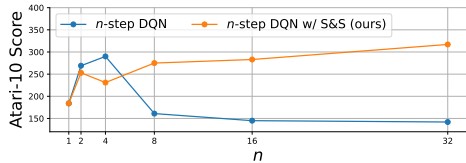

(a) Retrace($\lambda$) gains limited performance improvements as maximum bootstrapping steps $N$ increases.

(b) $n$-step DQN is sensitive to the choice of $n$. Varying $n$ lead to performance instability.

Figure 1: (a) Performance of Retrace($\lambda$) with varying $N$. Retrace($\lambda$) w/ S&S denotes EMB($\lambda$) as formally defined in Section 4.3. (b) Performance of $n$-step DQN with varying $n$. $n$-step DQN w/ S&S selects the bootstrapping step that maximizes $b(Z_n)$ (Equation (4)). Implementation details are provided in Appendix B and Appendix F.

as shown in Figure 1(a), the performance of Retrace($\lambda$) plateaus at $N = 8$, indicating that it struggles to propagate critical temporal information from salient bootstrapping steps, thereby limiting potential performance gains. Salience is a fundamental concept and mechanism in neuroscience that prioritizes the most informative and relevant signals in attention, decision-making, and learning (Rumbaugh et al., 2007; Fecteau & Munoz, 2006). Moreover, as illustrated in Figure 1(b), the choice of $n$ in $n$-step DQN significantly affects the performance stability, often requiring manual tuning of proper $n$ (Hessel et al., 2018; Yarats et al., 2021). The above observations show a central challenge remaining in existing multi-step RL methods: *Which bootstrapping steps are salient and stable for Multi-step RL?* **While learning with maximal values among different bootstrapping steps accelerates the propagation of critical temporal information (Wright et al., 2013; He et al., 2016; Wang et al., 2024b), directly bootstrapping on these salient value estimation risks Q-value explosion and overestimation, leading to instability which is closely related to large TD residuals (Baird et al., 1995; Durugkar & Stone, 2018; Parisi et al., 2019).**

Motivated by these insights, this paper investigates how to advance multi-step RL by learning from salient and stable bootstrapping steps, thereby *harnessing the benefits of maximal value propagation while mitigating the adverse effects of large TD residuals*. We analyze salience and stability (S&S) in multi-step RL and introduce a novel metric to qualify them. Our motivation experiments demonstrate that integrating this novel metric efficiently advances existing multi-step RL methods. As shown in Figure 1(a), it enables Retrace($\lambda$) to achieve consistent performance gains with increasing $N$. It substantially reduces the sensitivity to $n$ of $n$-step DQN, yielding stable performance improvements as $n$ increases, as illustrated in Figure 1(b). Based on these observations, we propose Expectation-Maximization Bootstrapping (EMB), which frames multi-step RL within the EM procedure (Dempster et al., 1977b) to enable principled learning of bootstrapping steps that balance S&S. Specifically, EMB alternates between two steps as follows:

E-step: estimates expectations under the posteriors to quantify the S&S of bootstrapping steps.

M-step: leverages these estimated expectations to guide the selection of bootstrapping steps.

Consequently, we present EMB($\lambda$) via applying this technique in the return-based Bellman operator, allowing the agent to efficiently learn from salient and stable bootstrapping steps while trading off the bias and variance. We also discuss the convergence and optimality properties of EMB($\lambda$). Empirical results on the Atari benchmark show that EMB($\lambda$) significantly outperforms existing multi-step RL baselines with superior sample efficiency and performance, achieving comparable performance to Retrace($\lambda$) with approximately $50\%$ fewer samples on the Atari-10 suite. Overall, our main contributions can be summarized as follows:

- We analyze the salience and stability (S&S) in multi-step RL and propose a novel metric to quantify the S&S of the bootstrapping steps.
- We propose EMB, a novel framework via formulating multi-step RL as the EM procedure.
- We present EMB($\lambda$) via integrating EMB into the return-based Bellman operator, allowing for efficient and stable learning that trades off bias and variance.
- We empirically demonstrate that our EMB($\lambda$) significantly outperforms existing multi-step RL baselines on the Atari Learning Environment benchmark.

## 2 RELATED WORKS

### 2.1 MULTI-STEP RL

Multi-step RL (Sutton & Barto, 2018) techniques aim to leverage temporal information from multiple future time steps to update value functions or policies. For example, N-step SARSA and N-step Q-learning estimate value functions using N-step returns rather than the standard one-step return (Sutton & Barto, 2018). Learning with the multi-step TD target is a simple yet efficient technique for improving learning efficiency, commonly used in A3C (Mnih et al., 2016) and Rainbow (Hessel et al., 2018). Furthermore, the multi-step learning technique can efficiently enhance the performance in RL with delayed feedback (Bouteiller et al., 2020; Wu et al., 2024b; 2025). For instance, to address temporal credit assignment in delayed reward settings, RUDDER (Arjona-Medina et al., 2019) employs the LSTM (Hochreiter & Schmidhuber, 1997) to decompose returns and perform reward redistribution. Highway RL (Wang et al., 2024b) proposes the highway gate operator to propagate temporal information from the large bootstrapping steps efficiently, thus improving the performance in RL tasks with delayed reward and large-scale navigation (Wang et al., 2024a; 2025). The $\lambda$-return approach utilizes eligibility traces to trade off the variance of Monte Carlo methods and the bias of one-step TD learning. In the off-policy setting, several extensions of TD($\lambda$) have been proposed to handle discrepancies between the behavior and target policies. Importance sampling (IS) (Precup et al., 2000a) reweights multi-step returns using corresponding policy ratios, always leading to amplified variance and performance degeneration. Tree-backup (TB($\lambda$)) (Precup et al., 2000b) mitigates the variance issue by scaling traces with policy probabilities. Q($\lambda$) (Harutyunyan et al., 2016) applies a constant weighting to multi-step returns, given the difference between behavior and target policies is small, and it is still commonly used in the current works (Daley & Amato, 2019; Gallici et al., 2025). Retrace($\lambda$) (Munos et al., 2016) modifies the traces via truncating policy ratios, guaranteeing stability and efficient learning. However, these existing multi-step RL methods still suffer from learning instability and inefficiency. To address these limitations, we propose a novel optimization framework for multi-step RL that adaptively assigns greater weight to salient multi-step temporal-difference targets, thereby enhancing both the stability and efficiency of learning.

### 2.2 EXPECTATION-MAXIMIZATION IN RL

Expectation-Maximization (EM) (Dempster et al., 1977a) is used to find maximum likelihood estimates of models with unobserved latent variables. Specifically, EM alternatively performs two steps: (1) the E-step, estimating the expected values of the latent variables based on the current parameters, and (2) the M-step, updating the parameters to maximize the likelihood of the data based on those estimated results. EM has been adapted to RL by reformulating RL as an inference problem (Ramachandran & Amir, 2007; Levine, 2018), allowing for the application of extensive and relevant optimization techniques to improve learning efficiency. For instance, CVPO (Liu et al., 2022) applies the variational RL technique to safe RL, enhancing the performance in the safety-critical scenarios. Utilizing a similar framework, VDPO (Wu et al., 2024a) extends the variational RL technique in addressing the delayed RL problem. These optimization approaches can also be applied in the policy search approach by integrating various projection techniques like VIP (Neumann, 2011). EM can be applied in the off-policy and entropy-regularized RL methods (MPO (Abdolmaleki et al., 2018a;b)) or actor–critic method (VIREL (Fellows et al., 2019)). Unlike these existing EM approaches, our main contribution is a novel optimization framework that reformulates multi-step RL through the lens of EM, providing a new pathway to analyze and improve multi-step methods.

## 3 PRELIMINARIES

**MDP and RL.** The RL problem is typically formalized as a Markov Decision Process (MDP) denoted by the tuple $\langle \mathcal{S}, \mathcal{A}, \mathcal{P}, \mathcal{R}, \gamma \rangle$. Here, $\mathcal{S}$ and $\mathcal{A}$ denote the finite state and action spaces, respectively. The transition function $\mathcal{P} : \mathcal{S} \times \mathcal{A} \to \Delta(\mathcal{S})$ specifies a probability distribution over the next state for each state-action pair, where $\Delta(\mathcal{S})$ represents the set of all probability distributions over the state space $\mathcal{S}$. The reward function $\mathcal{R} : \mathcal{S} \times \mathcal{A} \to [-R_{max}, R_{max}]$ is the bounded, and $\gamma \in [0, 1)$ is the discount factor. At each timestep $t$, the agent observes the state $s_t$ and select the action $a_t \sim \pi(\cdot|s_t)$ based on the policy $\pi : \mathcal{S} \to \Delta(\mathcal{A})$, then receive the reward $r_t := \mathcal{R}(s_t, a_t)$ and the new state $s_{t+1} \sim \mathcal{P}(\cdot|s_t, a_t)$. The agent's objective is to find the optimal policy $\pi^*$ which can maximize

the discounted return $\mathcal{J}(\pi) := \mathbb{E}_{\tau \sim p_\pi} [\sum_{t=0}^{\infty} \gamma^t r_t]$, where $\tau = \{s_t, a_t\}_{t=0}^{\infty}$ is the trajectory induced by $\pi$, sampling from the trajectory distribution $p_\pi$. For the policy $\pi$, its corresponding Q-function is defined as $Q^\pi(s, a) := \mathbb{E}_{\tau \sim p_\pi} [\sum_{t=0}^{\infty} \gamma^t r_t | s_0 = s, a_0 = a]$.

**TD Learning and Bellman Optimality Operator.** TD learning enables the agent to update the Q-function based on the current estimation of the future return, rather than waiting until the end of the episode. This bootstrapping property makes TD learning both more data-efficient and more suitable for online learning settings. The Bellman optimality operator $\mathcal{B}$ is defined as follows:

$$\mathcal{B}Q(s_t, a_t) := r_t + \gamma \max_a Q(s_{t+1}, a). \tag{1}$$

By iteratively applying $\mathcal{B}$, $Q$ can finally converge to the fixed point, which corresponds to the optimal Q-function $Q^*$.

**Multi-step RL and Eligibility traces.** Beyond the one-step TD learning (Equation (1)), multi-step RL accelerates the learning process by using the $n$-step TD target $Z_n$ defined as follows:

$$Z_n := \sum_{i=0}^{n-1} \gamma^i r_{t+i} + \gamma^n \max_a Q(s_{t+n}, a). \tag{2}$$

However, in the off-policy RL settings where the data is collected from a behavior policy $\mu$, the shorter bootstrapping steps $n$ may raise the bias issue, while the longer bootstrapping steps $n$ may raise the variance issue. To efficiently trade off the bias and variance, the return-based Bellman operator can address this issue via weighting the different multi-step TD targets with eligibility traces. Formally, the return-based Bellman operator is defined as

$$\mathcal{T}Q(s_t, a_t) := Q(s_t, a_t)$$
$$+ \mathbb{E}_{\{s_{t+i}, a_{t+i}, r_{t+i}\}_{i=0}^N \sim p_\mu} \left[ \sum_{i=0}^{N} \gamma^i (\Pi_{j=1}^i c_j)(r_{t+i} + \gamma \max_a Q(s_{t+i+1}, a) - Q(s_{t+i}, a_{t+i})) \right], \tag{3}$$

where $p_\mu$ is the trajectory distribution induced by the behavior policy $\mu$. Specifically, $c_j$ is the eligibility trace coefficient used to control the contribution of off-policy updates. For instance, Retrace($\lambda$) defines $c_j = \lambda \min(1, \frac{\pi(a_{t+j}|s_{t+j})}{\mu(a_{t+j}|s_{t+j})})$, where $\lambda \in [0, 1]$ is the trace decay parameter.

## 4 EXPECTATION-MAXIMIZATION BOOTSTRAPPING

In this section, we first analyze the salience and stability in multi-step RL (Section 4.1). Based on this, we introduce Expectation-Maximization Bootstrapping (EMB) by reformulating multi-step RL as the EM procedure in Section 4.2. In Section 4.3, we present EMB($\lambda$) by applying EMB to the return-based Bellman operator, which enables efficient learning from the salient and stable bootstrapping steps while trading off the bias and variance. We also conclude the discussion of the convergence and optimality guarantees.

### 4.1 SALIENCE AND STABILITY ANALYSIS

In multi-step RL, leveraging the maximal value can accelerate the temporal information propagation across long temporal distances (Wright et al., 2013; He et al., 2016; Wang et al., 2024b). For instance, Highway RL (Wang et al., 2024b) extracts critical information from bootstrapping steps via learning from the maximal multi-step TD target. However, directly learning this maximal target is prone to Q-value explosion and overestimation, thus destabilizing the learning process. This instability issue is further reflected in the large TD residuals, as investigated in prior studies (Baird et al., 1995; Durugkar & Stone, 2018; Parisi et al., 2019). Consequently, the central challenge in multi-step RL is to exploit salient multi-step TD targets (salience) while mitigating the adverse effects of large TD residuals (stability).

Building on these observations, we introduce a novel metric $b(Z_n) \in [0, 1]$ to quantify the salience and stability (S&S) of $Z_n$ as follows:

$$b(Z_n) := \underbrace{\frac{Z_n - \min_{j \in [1,N]} Z_j}{\max_{j \in [1,N]} Z_j - \min_{j \in [1,N]} Z_j}}_{\overline{Z_n}} * \left( 1 - \underbrace{\frac{\delta_n - \min_{j \in [1,N]} \delta_j}{\max_{j \in [1,N]} \delta_j - \min_{j \in [1,N]} \delta_j}}_{\overline{\delta_n}} \right), \quad (4)$$

where $\delta_n := ||Q - Z_n||$ is the TD residual of $Z_n$. Specifically, $\overline{Z_n}$ and $1 - \overline{\delta_i}$ measure the S&S of the TD target $Z_n$, respectively. Multiplying these two terms, $b(Z_n)$, can measure how salient and stable $Z_n$ is. Furthermore, we provide the following justification for Equation (4):

1. $\overline{Z_n}$ measures the likelihood that $Z_n$ provides salient TD target for bootstrapping. Inspired by existing works (Wright et al., 2013; He et al., 2016; Wang et al., 2024b), learning with maximal value among different bootstrapping steps can enhance the learning efficiency.

2. $\overline{\delta_n}$ measures the likelihood that $Z_n$ leads to learning instability. Existing works (Baird et al., 1995; Durugkar & Stone, 2018; Parisi et al., 2019) show that learning instability is usually related to the large TD residual.

3. $b(Z_n)$ indicates the joint likelihood that $Z_n$ is both salient and stable. The design of Equation (4) is inspired by learnability (Tzannetos et al., 2023), which shares a similar definition in the context of unsupervised environment design (Rutherford et al., 2024).

As shown in Figure 1, we have empirically demonstrated that Equation (4) can efficiently advance existing multi-step RL methods, including $n$-step DQN and Retrace($\lambda$). Building on this result, we next incorporate Equation (4) into a novel optimization framework for multi-step RL.

## 4.2 MULTI-STEP RL AS EXPECTATION-MAXIMIZATION

In this section, we present Expectation-Maximization Bootstrapping (EMB), a novel optimization method for multi-step RL. The objective of the multi-step RL can be interpreted as finding a latent model parameterized by $\theta$ that maximizes the log-evidence of the updated Q-function $Q_{\text{new}}$: $\max_\theta \log p(O|Q; \theta)$, where $O$ denotes the event of the optimality (e.g., $Q_{\text{new}}$ is $Q^*$). Specifically, $\theta$ specifies the distribution of the bootstrapping steps, which determines the corresponding multi-step TD targets. Therefore, we have

$$p(O|Q; \theta) = p(O|Z_n, Q)p(Z_n|Q; \theta),$$

where $p(O|Z_n, Q)$ is the likelihood of the optimality given $Z_n$ and $Q$, and $p(Z_n|Q; \theta)$ is the latent distribution over multi-step TD targets given $Q$. Then, we can introduce the variational distribution $q(Z_n|O, Q)$ and leverage Jensen's inequality to derive the lower bound $L(\theta, q)$:

$$\log p(O|Q; \theta) \geq \underbrace{\mathop{\mathbb{E}}_{Z_n \sim q(Z_n|O,Q)} \log \left[ \frac{p(O|Z_n, Q)p(Z_n|Q; \theta)}{q(Z_n|O, Q)} \right]}_{L(\theta, q)}. \quad (5)$$

The derivation of Equation (5) can be found in Appendix C. We adopt the EM approach to maximize Equation (5), alternating between the E-step and M-step. Specifically, E-step infers the variational distribution $q$; and M-step updates $\theta$ via maximizing Equation (5) with respect to $p(Z_n|Q; \theta)$.

**E-step.** In the $k$-th optimization, we have fixed $p(Z_n|Q; \theta^k)$ in Equation (5), the objective of E-step becomes as follows:

$$q^{k+1}(Z_n|O, Q) = \arg\max_q L(\theta^k, q) = \arg\min_q \text{KL}\left( q(Z_n|O, Q) || p(O|Z_n, Q)p(Z_n|Q; \theta^k) \right), \quad (6)$$

where KL is Kullback–Leibler divergence. Equation (6) implies that $q^{k+1}(Z_n|O, Q)$ needs to match the $p(O|Z_n, Q)p(Z_n|Q; \theta^k)$ as follows:

$$q^{k+1}(Z_n|O, Q) \propto p(O|Z_n, Q)p(Z_n|Q; \theta^k). \quad (7)$$

Specifically, $p(O|Z_n, Q)$ is estimated by applying Equation (4), which quantifies the S&S of $Z_n$. As discussed in Section 4.1, obtaining the optimality requires leveraging salient bootstrapping steps while mitigating the adverse effects of large TD residuals.

**M-step.** Based on Equation (7) in the previous E-step, the lower bound (Equation (5)) becomes:

$$L(\theta, q^{k+1}) \propto \mathop{\mathbb{E}}_{Z_n \sim p(Z_n|Q;\theta^k)} \left[ p(O|Z_n, Q) \log \left[ \frac{p(Z_n|Q;\theta)}{p(Z_n|Q;\theta^k)} \right] \right]. \tag{8}$$

The derivation of Equation (8) can be found in Appendix C. Ignoring the constant term in Equation (8), the objective of M-step becomes as follows:

$$\theta^{k+1} = \arg\max_{\theta} \mathop{\mathbb{E}}_{Z_n \sim p(Z_n|Q;\theta^k)} \left[ p(O|Z_n, Q) \log p(Z_n|Q;\theta) \right], \tag{9}$$

showing that $p(Z_n|Q;\theta^{k+1})$ should match $p(O|Z_n, Q)$ estimated by Equation (4). Then, we can update $Q$ based on the $Z_n \sim p(Z_n|Q;\theta^{k+1})$. For instance, we can select the bootstrapping steps with maximal $b(Z_n)$, i.e., $\arg\max_{n \in [1,N]} b(Z_n)$.

In summary, we have successfully formulated the multi-step RL as the EM approach alternating between E-step (Equation (7)) and M-step (Equation (9)), which enables the learning on the salient and stable bootstrapping steps. In the next section, we will incorporate EMB with the return-based Bellman operator to effectively balance the bias-variance trade-off in multi-step RL.

### 4.3 EMB($\lambda$)

---

**Algorithm 1** Expectation-Maximization Bootstrapping (EMB ($\lambda$))

---

**Initialize:** Q-function $Q$; $\epsilon$ for exploration;
**for** Each update step **do**
    Collect $N$-step transition $\{s_{t+i}, a_{t+i}, r_{t+i}\}_{i=0}^N$ using $Q$ with $\epsilon$-greedy policy $\mu$
    Calculate multi-step TD targets $\{Z_n\}_{n=1}^N$ via Equation (2)
    Calculate $\{b(Z_n)\}_{n=1}^N$ via Equation (4) # E-step
    Update $Q$ via Equation (10) # M-step
**end for**
**Output:** $Q$

---

In this section, we present EMB($\lambda$) (Algorithm 1) by applying our EMB technique in the return-based Bellman operator Equation (3) via replacing the traces of Retrace($\lambda$) with $b(Z_n)$-weighted traces. Formally, we present the EMB return-based Bellman operator $\mathcal{T}_{\text{EMB}}$ defined as follows:

$$\mathcal{T}_{\text{EMB}} Q(s_t, a_t) := Q(s_t, a_t)$$
$$+ \mathop{\mathbb{E}}_{\{s_{t+i}, a_{t+i}, r_{t+i}\}_{i=0}^N \sim p_\mu} \left[ \sum_{i=0}^N \gamma^i (\Pi_{j=1}^i b_j)(r_{t+i} + \gamma \max_a Q(s_{t+i+1}, a) - Q(s_{t+i}, a_{t+i})) \right], \tag{10}$$

where $b_j := \lambda \, b(Z_j) \min\left(1, \frac{\pi(a_j|s_j)}{\mu(a_j|s_j)}\right)$. EMB($\lambda$) can efficiently learn from the salient and stable bootstrapping steps while trading off the bias and variance. Next, based on the theoretical foundation (Lemma 4.1 and Lemma 4.2) established by Retrace($\lambda$) (Munos et al., 2016), we discuss the theoretical properties of EMB($\lambda$) in terms of optimality and convergence (Proposition 4.1). This analysis is conducted under the same assumptions and tabular setting as in Retrace($\lambda$) (Munos et al., 2016).

**Lemma 4.1 (Contraction property of Equation (3), Theorem 1 in (Munos et al., 2016))** *The return-based Bellman operator (Equation (3)) has a unique fixed point $Q^\pi$. For any $Q$, we have* $||\mathcal{T}Q - Q^\pi|| \le \gamma ||Q - Q^\pi||$, *if $c_j \in \left[0, \frac{\pi(a_j|s_j)}{\mu(a_j|s_j)}\right]$.*

**Lemma 4.2 (Optimality property of Equation (3), Theorem 2 in (Munos et al., 2016))**
*Consider the sequence of behavior policies $\{\mu_k\}$, target policies $\{\pi_k\}$, the return-based Bellman operators $\{\mathcal{T}_k\}$, and the Q-function $Q_{k+1} = \mathcal{T}_k Q_k$ where $c_j \in \left[0, \frac{\pi_k(a_j|s_j)}{\mu_k(a_j|s_j)}\right]$. Assume $\epsilon_k$ is the difference between $\pi_k$ and the greedy policy induced by $Q_k$. We have $||Q_{k+1} - Q^*|| \le \gamma ||Q_k - Q^*|| + \epsilon_k ||Q_k||$, then $Q_k \to Q^*$ if $\epsilon_k \to 0$.*

**Proposition 4.1 (Contraction and optimality of Equation (10))** *EMB return-based Bellman operator (Equation (10)) has the consistent contraction and optimality properties as illustrated in Lemma 4.1 and Lemma 4.2, respectively.*

The proof of Proposition 4.1 is straightforward via following the proof sketch of (Munos et al., 2016) since we have $b_j \in \left[0, \frac{\pi(a_j|s_j)}{\mu(a_j|s_j)}\right]$. Proposition 4.1 demonstrated that, under this modification, our operator (Equation (10)) preserves these theoretical guarantees. We have to clarify that this result is an adaptation of the existing theoretical foundation of Retrace($\lambda$), rather than our main contribution.

## 5 EXPERIMENTS

### 5.1 EXPERIMENTAL SETTINGS

**Benchmark.** In this work, we adopt the Atari Learning Environment (ALE) (Bellemare et al., 2013) as our evaluation benchmark, following common practice in the literature. Specifically, we adopt the Atari-10 and Atari-57 suites to ensure consistency with existing studies. For the evaluation metric, we adopt the commonly-used human-normalized score: $\frac{R_{alg} - R_{random}}{R_{human} - R_{random}}$, where $R_{human}$ and $R_{random}$ are scores of human and random players, respectively.

**Baselines.** For the baselines, we compare our EMB($\lambda$) with the popular and SOTA multi-step RL methods, including $n$-step (Sutton & Barto, 2018), Highway-step (Wang et al., 2024b), Importance Sampling (IS) (Precup et al., 2000a), TB($\lambda$) (Precup et al., 2000b), Q($\lambda$) (Harutyunyan et al., 2016), and Retrace($\lambda$) (Munos et al., 2016). For each method, the key hyperparameters are tuned according to the performance on the Atari-10 suite. Specifically, for $n$-step and Highway-step, we find the maximum bootstrapping steps $n \in \{2, 4, 8\}$. For TB($\lambda$), Q($\lambda$), Retrace($\lambda$), and our EMB($\lambda$), we find $\lambda \in \{0.2, 0.4, 0.6, 0.8\}$. Specifically, Q($\lambda$) can be regarded as PQN (Gallici et al., 2025), and we follow the original paper by setting $\lambda = 0.65$.

**Implementation.** For practical implementation, we adopt EnvPool (Weng et al., 2022), which provides a vectorized ALE and significantly accelerates training speed. Following the existing literature, all methods are run over 3 random seeds with 200M total frames on ALE. To efficiently conduct a comprehensive comparison between existing multi-step RL baselines and our EMB($\lambda$), we select PQN (Gallici et al., 2025) as the base algorithm, the vectorized variant of Deep Q-Network implemented by JAX (Bradbury et al., 2018), given its fast training speed, low memory requirement, and stable performance guarantee. Thus, the average training time on NVIDIA GeForce RTX 4090 is around 1 hour. We mainly adopt the hyperparameter setting from PQN (Gallici et al., 2025), and the specific implementation detail can be found in Appendix B. The codebase for reproducing our experimental results can be found in the supplementary material.

### 5.2 EXPERIMENTAL RESULTS

#### 5.2.1 PERFORMANCE COMPARISON

We first evaluate the performance on the Atari-10 suite. As summarized in Table 1, EMB($\lambda$) significantly outperforms the other baselines, achieving the Atari-10 score of $581$ and surpassing Retrace($\lambda$) ($343$), the best baseline, by $69\%$. On the Atari-57 suite, we report the human-normalized scores (Median and Mean), interquartile mean (IQM) (Agarwal et al., 2021), Optimality Gap, and better than human (>Human). As shown in Table 1, Q($\lambda$) and Retrace($\lambda$) exhibit higher mean scores, primarily due to human-normalized scores of a few games dominating the average, which also leads to large variance (e.g., Asterix-v5 and VideoPinball-v5). Nevertheless, EMB($\lambda$) achieves the highest median score (2.15), demonstrating consistently strong performance across the Atari-57 suite. EMB($\lambda$) achieves the highest IQM score (2.86), significantly outperforming the best baseline (2.73). Furthermore, EMB($\lambda$) can achieve the human-level performance on 43 out of 57 Atari games, compared to 41 Atari games achieved by the best baseline. Overall, these empirical results demonstrate that EMB($\lambda$) can achieve superior performance compared to existing baselines across various Atari games. We also provided the ALE scores on the Atari-57 suite in Appendix D. The performance on the Atari-10 suite with 5 random seeds and 400M frames are provided in Appendix H and Appendix I, respectively.

Table 1: Performance on Atari Learning Environment. The best performance is highlighted.

| Method | Atari-10 Score ↑ | Atari-57 | | | | | |
| --- | --- | --- | --- | --- | --- | --- | --- |
| | | Median ↑ | Mean ↑ | IQM ↑ | Optimality Gap ↓ | >Human ↑ | Best Count ↑ |
| One-step | 184 | 1.12 | 9.73 | 1.23 | 0.32 | 30 | 2 |
| $n$-step | 290 | 1.50 | 13.15 | 2.40 | 0.22 | 38 | 7 |
| Highway-step | 246 | 1.19 | 9.28 | 1.62 | 0.26 | 33 | 6 |
| IS | 140 | 0.74 | 2.36 | 0.81 | 0.38 | 24 | 4 |
| TB($\lambda$) | 211 | 1.14 | 10.57 | 1.53 | 0.29 | 31 | 4 |
| Q($\lambda$) | 322 | 1.89 | 13.62 | 2.63 | 0.20 | 39 | 11 |
| Retrace($\lambda$) | 343 | 1.68 | **14.22** | 2.73 | 0.20 | 41 | 21 |
| EMB($\lambda$) (ours) | **581** | **2.15** | 13.49 | **2.86** | **0.15** | **43** | **27** |

### 5.2.2 LEARNING EFFICIENCY

As shown in Figure 2, we further analyse the learning curves on the Atari-10 suite. Specifically, EMB($\lambda$) significantly outperforms the best baseline in several games, including Amidar-v5, DoubleDunk-v5, Frostbite-v5, and Phoenix-v5. EMB($\lambda$) can achieve performance comparable to the best baselines on some games, such as BattleZone-v5, Bowling-v5, Qbert-v5, and Riverraid-v5. There are also a few games, such as KungfuMaster-v5 and NameThisGame-v5, where EMB($\lambda$) underperforms compared to specific baselines. Overall, EMB($\lambda$) show a superior performance in most of the games in the Atari-10 suite. Overall, as shown in Figure 3(a), EMB($\lambda$) consistently exhibits a leading learning efficiency on the Atari-10 suite. The learning curves demonstrate that EMB($\lambda$) not only achieve superior final performance, but also better sample efficiency throughout the entire learning process, achieving a performance similar to the best baseline with 50% less amount of samples (100M frames). We also provided the learning curves on the Atari-57 suite in Appendix E.

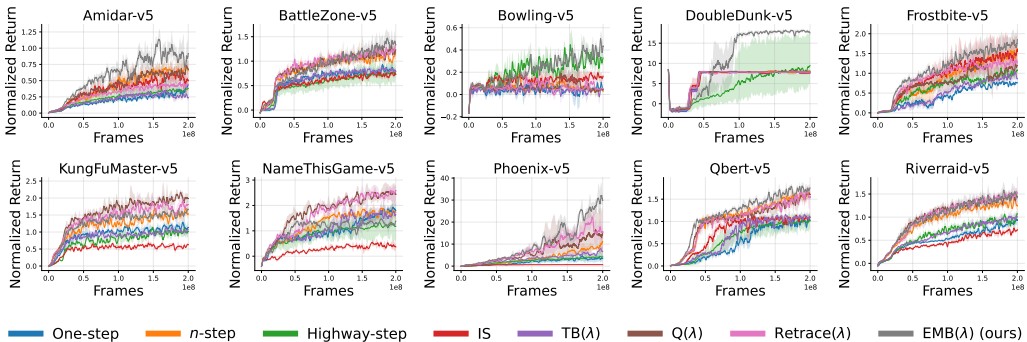

Figure 2: Learning curves on the Atari-10 suite.

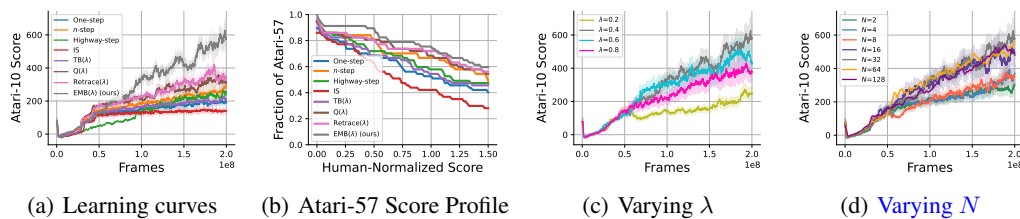

(a) Learning curves  (b) Atari-57 Score Profile  (c) Varying $\lambda$  (d) Varying $N$

Figure 3: (a) Learning curves on the Atari-10 suite. (b) The fraction of games exceeding the human-normalized score threshold on the Atari-57 suite. Ablation studies on varying (c) $\lambda$ and (d) $N$ of EMB($\lambda$).

### 5.2.3 ATARI-57 SCORE PROFILE

We present the score profile on the Atari-57 suite in Figure 3(b), illustrating the fraction of games exceeding different human-normalized score thresholds. EMB($\lambda$) maintains the highest fraction of games across different thresholds, consistently outperforming the baselines. Specifically, EMB($\lambda$) can achieve 0.5 of human-normalized score on nearly 90% of games, compared to 81% of the best baseline. For 1.0 of human-normalized score, EMB($\lambda$) surpasses the human-level player on around

75% of games while the best baseline achieves approximately 72% of games. When the threshold increases to 1.5, EMB($\lambda$) can achieve around 60% of games, higher than 56% achieved by the best baseline. These consistent improvements across different thresholds of hunman-normalized score demonstrate that EMB($\lambda$) can achieve better performance across various games generally.

### 5.2.4 ABLATION STUDIES

We investigate the influence of $\lambda$ in EMB($\lambda$) and conduct the ablation studies with $\lambda \in \{0.2, 0.4, 0.6, 0.8\}$ on the Atari-10 suite. As presented in Figure 3(c), $\lambda = 0.4$ achieves the strongest performance while maintaining stable learning. Specifically, low value ($\lambda = 0.2$) tends to utilize the short-step TD targets, which degenerates the learning efficiency, whereas $\lambda = 0.6$ and $\lambda = 0.8$ tend to leverage the long-step TD returns with high variance, resulting in limited performance. As shown in Figure 3(d), we also conduct ablation studies with bootstrapping steps $N \in \{2, 4, 8, 16, 32, 64, 128\}$ on the Atari-10 suite. The results demonstrate that EMB($\lambda$) show that performance saturates at $N = 32$ and shows a slight decline at $N = 64$ and $N = 128$. We present the per-game sensitivity analysis for NameThisGame-v5 and KungFuMaster-v5 in Table 4.

### 5.3 LIMITATIONS AND FUTURE WORKS

**Computational Complexity.** EMB($\lambda$) requires computing $\{b(Z_n)\}_{n=1}^N$ and the corresponding importance sampling ratios at every timestep. Therefore, EMB($\lambda$) cannot take advantage of the more efficient incremental techniques (Gallici et al., 2025), resulting in higher computational cost.

**On-policy RL Methods.** EMB($\lambda$) is an off-policy multi-step RL method. An interesting direction for future work is to investigate how the EMB framework can be adapted and integrated with on-policy methods such as Schulman et al. (2017; 2015).

**Challenging Tasks.** We also observe that sparse reward games (e.g., MontezumaRevenge-v5) remain a significant challenge for EMB($\lambda$). It is therefore necessary to explore how exploration strategies (Badia et al., 2020b; Oh et al., 2018; Ecoffet et al., 2019) can be effectively integrated with EMB($\lambda$). These challenges will be investigated in our future work.

## 6 CONCLUSION

In this work, we investigate which bootstrapping steps are salient and stable, a central challenge remaining in existing multi-step RL methods. We first analyze the salience and stability in multi-step RL and introduce a novel metric to quantify the bootstrapping steps. Then, we propose EMB, a novel optimization framework via reformulating multi-step RL as the Expectation-Maximization procedure. EMB alternates between(1) E-step: estimating the likelihood that the bootstrapping steps are both salient and stable; (2) M-step: leveraging the estimated likelihoods to select bootstrapping steps. Applying the EMB approach in the return-based Bellman operator, we propose EMB($\lambda$), which can efficiently learning from the salient and stable bootstrapping steps while trading off the bias and variance. We also theoretically discuss the convergence and optimality properties of EMB($\lambda$). On the Atari Learning Environment benchmark, we empirically show that our EMB($\lambda$) effectively outperforms existing multi-step RL baselines with superior performance and sample efficiency.

## ETHICS STATEMENT

We affirm that all authors have read and adhere to the ICLR Code of Ethics. This work does not involve human or animal subjects, sensitive personal data, or privacy risks. The use of LLMs was limited to writing support and language refinement. LLMs were not used in the design of algorithms, the development of theoretical results, or the execution of experiments, ensuring that all core scientific contributions are entirely the work of the authors.

## REPRODUCIBILITY STATEMENT

We provide the implementation detail in Appendix B. We also provide the codebase reproducing our experimental results in the supplementary material.

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

## A   LLM Usage Statement

In this work, the use of LLMs was restricted to writing support and language refinement. Specifically, LLMs assisted in enhancing the clarity and coherence of the manuscript. LLMs were not used in the design of algorithms, the development of theoretical results, or the execution of experiments, ensuring that all core scientific contributions are entirely the work of the authors.

## B   Implementation Detail

Our implementation is based on PQN (Gallici et al., 2025), and we also provide the codebase for reproducing our experimental results in the supplementary material. We mainly adopt the hyperparameter settings from PQN (Gallici et al., 2025), and the specific hyperparameter settings are detailed in Table 2.

Table 2: Hyperparameter settings.

| Hyperparameter | Value |
| --- | --- |
| Total Frames | 200M |
| Num of Environments | 128 |
| Num of Steps | 32 |
| Initial $\epsilon$ | 1.0 |
| Final $\epsilon$ | 1e-3 |
| Decay Factor of $\epsilon$ | 1e-1 |
| Epoch | 2 |
| Minibatch Size | 32 |
| Learning Rate | 2.5e-4 |
| Max Grad Norm | 10 |
| Discount Factor $\gamma$ | 0.99 |
| Maximum Bootstrapping Steps $N$ | 2 (Highway-step), 4 ($n$-step), 32 (Others) |
| $\lambda$ of Traces | 0.4 (EMB($\lambda$)), 0.65 (Q($\lambda$)), 0.6 (Others) |

## C  EQUATION DERIVATION

We provide the derivation of Equation (5) and Equation (8) as follows.

### C.1  EQUATION (5)

$$
\begin{aligned}
\log p(O|Q;\theta) &= \log\left[p(O|Z_n,Q)p(Z_n|Q;\theta)\right], \\
&= \log \mathop{\mathbb{E}}_{Z_n\sim q(Z_n|O,Q)}\left[\frac{p(O|Z_n,Q)p(Z_n|Q;\theta)}{q(Z_n|O,Q)}\right], \\
&\geq \mathop{\mathbb{E}}_{Z_n\sim q(Z_n|O,Q)}\log\left[\frac{p(O|Z_n,Q)p(Z_n|Q;\theta)}{q(Z_n|O,Q)}\right], \\
&= \mathop{\mathbb{E}}_{Z_n\sim q(Z_n|O,Q)}\log\left[p(O|Z_n,Q)\right] - \mathrm{KL}\left(q(Z_n|O,Q||p(Z_n|Q;\theta))\right), \\
&= L(\theta,q).
\end{aligned}
\tag{11}
$$

### C.2  EQUATION (8)

$$
\begin{aligned}
L(\theta,q^{k+1}) &= \mathop{\mathbb{E}}_{Z_n\sim q^{k+1}(Z_n|O,Q)}\log\left[\frac{p(O|Z_n,Q)p(Z_n|Q;\theta)}{q^{k+1}(Z_n|O,Q)}\right], \\
&\propto \sum_{Z_n}\left[p(Z_n|Q;\theta^k)p(O|Z_n,Q)\log\left[\frac{p(O|Z_n,Q)p(Z_n|Q;\theta)}{p(Z_n|Q;\theta^k)p(O|Z_n,Q)}\right]\right], \\
&= \mathop{\mathbb{E}}_{Z_n\sim p(Z_n|Q;\theta^k)}\left[p(O|Z_n,Q)\log\left[\frac{p(Z_n|Q;\theta)}{p(Z_n|Q;\theta^k)}\right]\right].
\end{aligned}
\tag{12}
$$

# D  ATARI LEARNING ENVIRONMENT SCORE

| Game | One-step | $n$-step | Highway-step | IS | TB($\lambda$) | Q($\lambda$) | Retrace($\lambda$) | EMB($\lambda$) (ours) |
|---|---|---|---|---|---|---|---|---|
| Alien-v5 | 1630.83 | 4369.17 | 1733.09 | 2704.32 | 1996.41 | 3024.17 | 3995.83 | **7439.58** |
| Amidar-v5 | 789.44 | 968.00 | 720.92 | 773.24 | 422.29 | 980.42 | 765.42 | **1535.71** |
| Assault-v5 | 7323.79 | 17652.29 | 3968.09 | 7921.17 | 8082.14 | **17667.63** | 16040.62 | 14630.75 |
| Asterix-v5 | 98880.60 | 311927.08 | 201310.68 | 4114.58 | 142710.42 | 338877.08 | **454095.83** | 223116.67 |
| Asteroids-v5 | 1967.50 | 1810.83 | **3261.34** | 565.39 | 1017.16 | 2395.00 | 2382.50 | 2125.30 |
| Atlantis-v5 | 12462.24 | 733025.00 | 660329.17 | 677562.50 | 326183.33 | 708141.67 | 803745.83 | **827379.17** |
| BankHeist-v5 | 1181.18 | **1510.42** | 1191.46 | 1203.61 | 1177.08 | 1415.83 | 1263.75 | 1320.83 |
| BattleZone-v5 | 30625.00 | 42221.35 | 32905.99 | 27375.00 | 29625.00 | 46833.33 | **50041.67** | 49333.33 |
| BeamRider-v5 | 2035.11 | 15152.00 | 10341.58 | 5509.00 | 2080.50 | 17051.83 | 17569.83 | **18027.25** |
| Berzerk-v5 | 740.83 | 2200.00 | 986.86 | 1241.12 | 715.48 | 2235.42 | 2897.50 | **8826.77** |
| Bowling-v5 | 32.18 | 31.29 | 29.17 | 47.21 | 28.79 | 30.04 | 30.19 | **83.54** |
| Boxing-v5 | 98.79 | **99.96** | 92.82 | 89.79 | 98.58 | 99.92 | 99.50 | 99.21 |
| Breakout-v5 | 451.38 | 432.92 | **558.60** | 168.25 | 427.62 | 492.83 | 451.58 | 418.00 |
| Centipede-v5 | 8926.12 | 6791.47 | **10647.58** | 2891.57 | 6139.04 | 8196.30 | 8158.04 | 7531.75 |
| ChopperCommand-v5 | 5597.92 | 7849.31 | 6015.66 | 8545.83 | 6112.50 | 14514.06 | 15279.17 | **21100.00** |
| CrazyClimber-v5 | 144166.67 | 154710.20 | 142113.96 | 126333.33 | 152691.67 | **174662.50** | 166175.00 | 155237.50 |
| Defender-v5 | 30756.25 | 51046.13 | 36606.59 | 10592.12 | 30208.33 | 53635.42 | **55968.75** | 45340.56 |
| DemonAttack-v5 | 115708.54 | 124742.88 | 71032.73 | 10415.83 | 123460.83 | 131077.29 | **131972.08** | 117816.46 |
| DoubleDunk-v5 | -1.08 | -0.75 | 10.70 | -1.92 | -1.67 | -1.33 | -1.33 | **20.00** |
| Enduro-v5 | 2302.50 | **2322.17** | 2245.33 | 1100.38 | 2261.83 | 2274.38 | 2316.21 | 2308.04 |
| FishingDerby-v5 | 25.48 | 37.74 | 23.61 | 16.92 | 28.88 | 47.08 | **48.92** | 36.38 |
| Freeway-v5 | 22.04 | 33.46 | 33.12 | 24.21 | 33.62 | **33.83** | **33.83** | 33.67 |
| Frostbite-v5 | 2975.00 | 6838.33 | 3263.39 | 7008.33 | 3639.43 | 5501.67 | 5814.18 | **7932.62** |
| Gopher-v5 | 22812.50 | 35223.74 | 13751.98 | 2015.00 | 21545.83 | 35937.50 | **44265.00** | 21969.17 |
| Gravitar-v5 | 368.75 | 797.59 | 605.50 | 1847.92 | 308.59 | 920.05 | 1002.08 | **1867.77** |
| Hero-v5 | 11376.57 | 16847.17 | 13980.76 | 24858.75 | 12848.33 | 20210.62 | 20245.21 | **39321.67** |
| IceHockey-v5 | -4.46 | -4.84 | -7.37 | -10.21 | -3.58 | -1.04 | **0.46** | -3.83 |
| Jamesbond-v5 | 400.00 | 2644.60 | 1128.23 | 754.17 | 850.00 | 1550.00 | 2410.42 | **4395.31** |
| Kangaroo-v5 | 11637.50 | 13441.98 | 7888.90 | 12145.83 | 9938.02 | **14041.67** | 13662.50 | 13420.83 |
| Krull-v5 | 8645.62 | 9916.30 | 8934.32 | 8596.67 | 8787.08 | **10115.00** | 9799.58 | 9535.42 |
| KungFuMaster-v5 | 25491.67 | 33591.67 | 27055.57 | 14366.67 | 24520.83 | **44044.92** | 41237.50 | 37275.00 |
| MontezumaRevenge-v5 | 0.00 | 0.00 | 0.00 | 0.00 | 0.00 | 0.00 | 0.00 | **16.67** |
| MsPacman-v5 | 2006.35 | 3964.00 | 2154.05 | 2395.00 | 2572.66 | 3948.33 | 3575.18 | **4348.33** |
| NameThisGame-v5 | 12830.42 | 12195.42 | 12378.31 | 4206.67 | 11889.58 | 16433.75 | **16770.00** | 10506.25 |
| Phoenix-v5 | 24872.92 | 61672.66 | 27650.89 | 5072.08 | 46751.25 | 89620.83 | 103607.92 | **193645.42** |
| Pitfall-v5 | -80.50 | -81.08 | -34.00 | **-27.71** | -70.96 | -32.58 | -80.88 | -33.29 |
| Pong-v5 | 20.46 | 20.98 | 20.55 | 20.96 | **21.00** | **21.00** | **21.00** | 20.88 |
| PrivateEye-v5 | -138.54 | 18.79 | -194.62 | 3.96 | 46.33 | -79.17 | **66.67** | 50.21 |
| Qbert-v5 | 13546.88 | 20378.12 | 14956.98 | 14754.17 | 14833.33 | 21198.96 | 22478.12 | **23419.27** |
| Riverraid-v5 | 16118.65 | 24622.50 | 16795.44 | 12854.17 | 17127.50 | 24372.92 | **25107.08** | 22402.50 |
| RoadRunner-v5 | 48416.67 | 147419.13 | 46172.33 | 43228.78 | 46273.05 | **149154.17** | 57760.68 | 106534.38 |
| Robotank-v5 | 55.00 | 65.33 | 61.32 | 18.25 | 57.62 | **72.42** | 71.96 | 67.48 |
| Seaquest-v5 | 15134.17 | 19827.71 | 18651.22 | 3229.17 | 19023.33 | 8084.17 | 12830.00 | **20815.83** |
| Skiing-v5 | -29974.88 | -29972.10 | -29978.30 | -30517.25 | -29974.12 | -29970.58 | -30512.67 | **-10920.38** |
| Solaris-v5 | 650.83 | 1684.70 | 1476.32 | 1051.67 | 1405.83 | **1913.33** | 1753.33 | 1773.62 |
| SpaceInvaders-v5 | 3097.08 | 7156.70 | 2969.97 | 848.87 | 3541.32 | 16096.25 | **17811.93** | 12070.62 |
| StarGunner-v5 | 142820.83 | 181217.68 | 95983.33 | 34416.67 | 165129.17 | 247129.17 | **256191.67** | 189012.50 |
| Surround-v5 | 4.50 | -1.56 | 0.88 | 0.50 | 2.33 | 5.79 | **7.66** | 7.58 |
| Tennis-v5 | -12.75 | -2.54 | 6.37 | -7.38 | -12.33 | -2.92 | -4.50 | **23.42** |
| TimePilot-v5 | 6364.06 | 12165.76 | 8629.69 | 3769.79 | 7120.83 | **12715.62** | 12354.43 | 12054.17 |
| Tutankham-v5 | 218.79 | 228.42 | 221.06 | 179.25 | 226.21 | 234.04 | 236.46 | **256.62** |
| UpNDown-v5 | 18826.25 | 163970.76 | 23037.27 | 15860.00 | 28183.75 | 207266.25 | **269195.42** | 255658.75 |
| Venture-v5 | 0.00 | 25.00 | 25.00 | 0.00 | 0.00 | 8.33 | 33.33 | **1016.67** |
| VideoPinball-v5 | 504708.17 | 545027.70 | 425270.66 | 11579.33 | 520307.21 | 525254.96 | **556346.71** | 425882.75 |
| WizardOfWor-v5 | 6837.50 | 13803.78 | 12366.50 | 2320.83 | 9906.90 | 20579.17 | **21829.17** | 18762.50 |
| YarsRevenge-v5 | 15588.62 | 114290.20 | 59684.58 | 72300.54 | 25560.33 | 109419.88 | 121101.08 | **141441.02** |
| Zaxxon-v5 | 10254.17 | 11064.70 | 7892.55 | 2857.29 | 8691.67 | 14387.50 | 13841.67 | **19718.75** |
| Best Count | 0 | 3 | 3 | 1 | 1 | 11 | 18 | **23** |

Table 3: Atari Learning Environment Scores. The best performance is highlighted.

# E LEARNING CURVES ON ATARI-57 SUITE

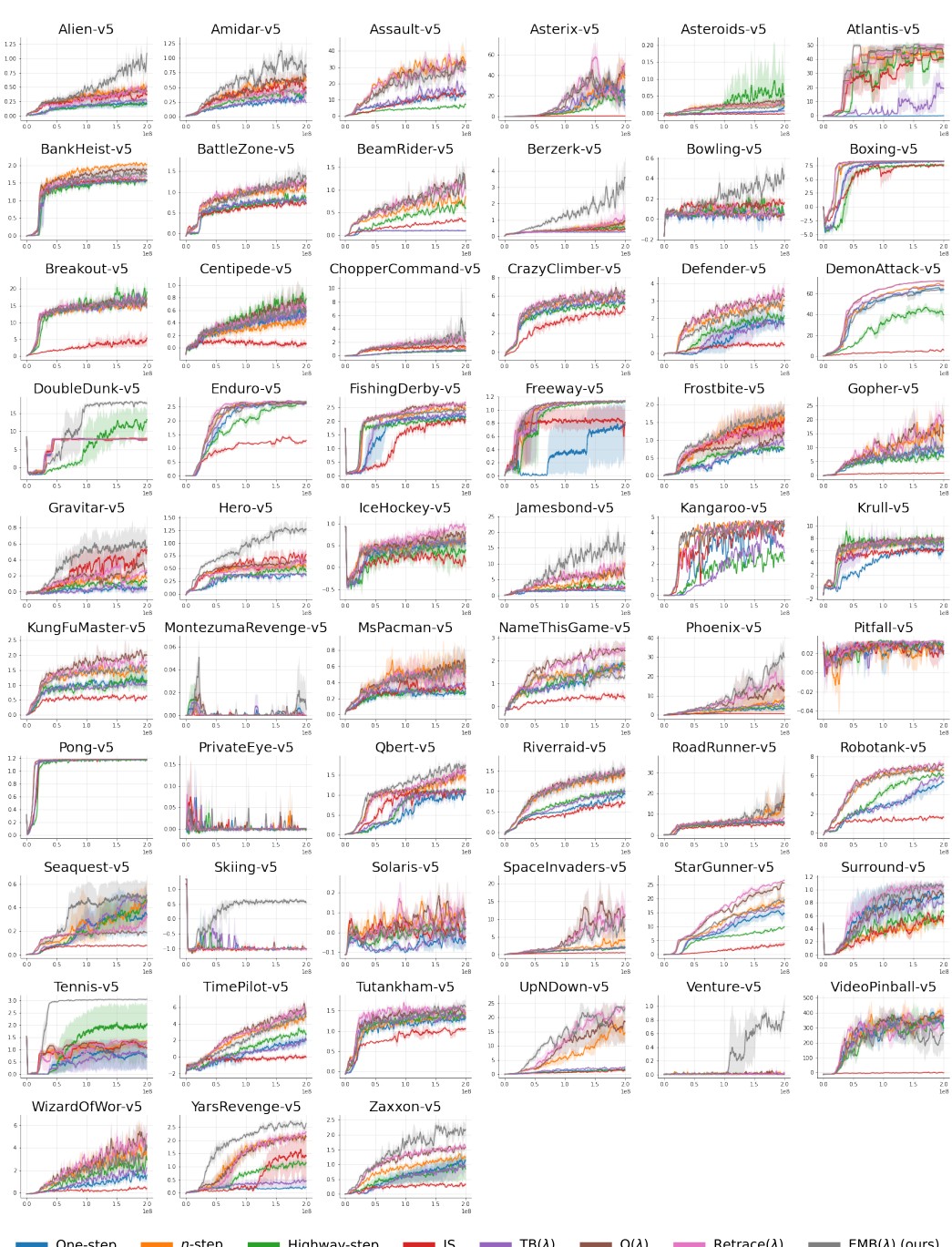

Figure 4: Learning curves on the Atari-57 suite.

## F  IMPLEMENTATION DETAIL OF MOTIVATION EXPERIMENT

For our motivation experiment (Figure 1), the hyperparameter settings remain the same as Table 2. Specifically, "Retrace($\lambda$) w/ S&S" illustrates the performance of the EMB($\lambda$). "$n$-step DQN w/ S&S" advances $n$-step DQN by selecting the single bootstrapping step that has the maximum $b(Z\_n)$ (e.g., $\arg\max_n b(Z_n)$).

## G  SENSITIVITY ANALYSIS OF EMB($\lambda$)'S PERFORMANCE TO $N$

As shown in Table 4, the results show that reducing the maximum bootstrapping steps $N$ significantly improves the performance of EMB($\lambda$). This confirms that the underperformance was not due to instability, but rather because the global hyperparameter setting allowed $b(Z\_n)$ to prioritize long-horizon updates in environments where shorter multi-step returns are more effective.

| $N$ | 2 | 4 | 8 | 16 | 32 | 64 | 128 |
|---|---|---|---|---|---|---|---|
| KungFuMaster-v5 | $1.63_{\pm 0.13}$ | $\mathbf{1.73_{\pm 0.02}}$ | $1.37_{\pm 0.25}$ | $1.44_{\pm 0.16}$ | $1.69_{\pm 0.08}$ | $1.50_{\pm 0.08}$ | $1.42_{\pm 0.12}$ |
| NameThisGame-v5 | $\mathbf{2.41_{\pm 0.06}}$ | $2.01_{\pm 0.11}$ | $1.92_{\pm 0.09}$ | $1.50_{\pm 0.19}$ | $1.35_{\pm 0.18}$ | $1.12_{\pm 0.15}$ | $1.23_{\pm 0.12}$ |

Table 4: Sensitivity of EMB($\lambda$)'s performance to $N$ on KungFuMaster-v5 and NameThisGame-v5, averaged across 3 random seeds.

## H  PERFORMANCE ON ATARI-10 SUITE (5 SEEDS)

We provide the performance on the Atari-10 suite in Table 5, which is consistent with our original findings and confirms the performance robustness of EMB($\lambda$).

| Game | One-step | $n$-step | Highway-step | IS | TB($\lambda$) | Q($\lambda$) | Retrace($\lambda$) | EMB($\lambda$) (ours) |
|---|---|---|---|---|---|---|---|---|
| Amidar-v5 | $0.37_{\pm 0.09}$ | $1.06_{\pm 0.14}$ | $0.31_{\pm 0.01}$ | $0.52_{\pm 0.14}$ | $0.25_{\pm 0.05}$ | $0.56_{\pm 0.04}$ | $0.53_{\pm 0.15}$ | $\mathbf{0.85_{\pm 0.20}}$ |
| BattleZone-v5 | $0.72_{\pm 0.06}$ | $1.06_{\pm 0.14}$ | $0.74_{\pm 0.06}$ | $0.71_{\pm 0.02}$ | $0.80_{\pm 0.06}$ | $\mathbf{1.26_{\pm 0.07}}$ | $1.25_{\pm 0.19}$ | $1.25_{\pm 0.19}$ |
| Bowling-v5 | $0.04_{\pm 0.01}$ | $0.06_{\pm 0.01}$ | $0.30_{\pm 0.06}$ | $0.17_{\pm 0.03}$ | $0.04_{\pm 0.02}$ | $0.07_{\pm 0.03}$ | $0.07_{\pm 0.03}$ | $\mathbf{0.37_{\pm 0.09}}$ |
| DoubleDunk-v5 | $7.88_{\pm 0.13}$ | $7.92_{\pm 0.39}$ | $10.25_{\pm 1.52}$ | $7.60_{\pm 0.12}$ | $7.71_{\pm 0.28}$ | $7.83_{\pm 0.09}$ | $7.81_{\pm 0.07}$ | $\mathbf{15.51_{\pm 2.67}}$ |
| Frostbite-v5 | $0.81_{\pm 0.07}$ | $1.64_{\pm 0.34}$ | $0.85_{\pm 0.08}$ | $1.57_{\pm 0.40}$ | $0.83_{\pm 0.08}$ | $1.34_{\pm 0.33}$ | $1.42_{\pm 0.17}$ | $\mathbf{1.93_{\pm 0.32}}$ |
| KungFuMaster-v5 | $1.07_{\pm 0.17}$ | $1.44_{\pm 0.07}$ | $1.09_{\pm 0.14}$ | $0.66_{\pm 0.08}$ | $1.15_{\pm 0.20}$ | $\mathbf{1.80_{\pm 0.35}}$ | $1.74_{\pm 0.25}$ | $1.53_{\pm 0.21}$ |
| NameThisGame-v5 | $1.58_{\pm 0.38}$ | $1.67_{\pm 0.31}$ | $1.17_{\pm 0.15}$ | $0.34_{\pm 0.08}$ | $1.79_{\pm 0.19}$ | $2.35_{\pm 0.31}$ | $\mathbf{2.47_{\pm 0.28}}$ | $1.34_{\pm 0.18}$ |
| Phoenix-v5 | $3.26_{\pm 1.38}$ | $5.93_{\pm 8.36}$ | $3.58_{\pm 0.49}$ | $0.66_{\pm 0.01}$ | $5.91_{\pm 2.24}$ | $12.43_{\pm 9.51}$ | $15.86_{\pm 3.09}$ | $\mathbf{23.70_{\pm 8.67}}$ |
| Qbert-v5 | $1.01_{\pm 0.06}$ | $1.46_{\pm 0.14}$ | $0.95_{\pm 0.07}$ | $1.09_{\pm 0.04}$ | $1.08_{\pm 0.08}$ | $1.51_{\pm 0.14}$ | $1.62_{\pm 0.15}$ | $\mathbf{1.76_{\pm 0.06}}$ |
| Riverraid-v5 | $0.86_{\pm 0.15}$ | $1.34_{\pm 0.17}$ | $0.86_{\pm 0.09}$ | $0.71_{\pm 0.04}$ | $0.89_{\pm 0.21}$ | $1.38_{\pm 0.15}$ | $\mathbf{1.44_{\pm 0.10}}$ | $1.41_{\pm 0.10}$ |

Table 5: Performance (mean $\pm$ standard deviation) on the Atari-10 suite over 200M frames, averaged across 5 random seeds.

## I  PERFORMANCE ON THE ATARI-10 SUITE WITH 400M TOTAL FRAMES

We provide the performance on the Atari-10 suite (400M frames) in Table 6, where EMB($\lambda$) achieves the highest performance on 7 out of 10 games.

| Game | One-step | $n$-step | Highway-step | IS | TB($\lambda$) | Q($\lambda$) | Retrace($\lambda$) | EMB($\lambda$) (ours) |
|---|---|---|---|---|---|---|---|---|
| Amidar-v5 | $0.66_{\pm 0.14}$ | $0.63_{\pm 0.05}$ | $0.63_{\pm 0.13}$ | $0.40_{\pm 0.02}$ | $0.39_{\pm 0.11}$ | $0.73_{\pm 0.03}$ | $0.78_{\pm 0.11}$ | $\mathbf{1.37_{\pm 0.16}}$ |
| BattleZone-v5 | $0.91_{\pm 0.07}$ | $0.87_{\pm 0.07}$ | $1.10_{\pm 0.02}$ | $0.73_{\pm 0.08}$ | $1.00_{\pm 0.22}$ | $1.50_{\pm 0.11}$ | $1.25_{\pm 0.11}$ | $\mathbf{1.53_{\pm 0.06}}$ |
| Bowling-v5 | $0.06_{\pm 0.02}$ | $0.26_{\pm 0.05}$ | $0.03_{\pm 0.01}$ | $0.27_{\pm 0.12}$ | $0.12_{\pm 0.10}$ | $0.04_{\pm 0.04}$ | $0.06_{\pm 0.00}$ | $\mathbf{0.32_{\pm 0.05}}$ |
| DoubleDunk-v5 | $8.00_{\pm 0.00}$ | $7.43_{\pm 0.25}$ | $7.77_{\pm 0.25}$ | $12.32_{\pm 5.32}$ | $7.62_{\pm 2.21}$ | $7.89_{\pm 0.25}$ | $7.92_{\pm 0.19}$ | $\mathbf{14.67_{\pm 5.20}}$ |
| Frostbite-v5 | $0.84_{\pm 0.07}$ | $2.13_{\pm 0.48}$ | $1.54_{\pm 0.44}$ | $1.17_{\pm 0.19}$ | $0.80_{\pm 0.05}$ | $1.54_{\pm 0.34}$ | $1.65_{\pm 0.53}$ | $\mathbf{2.29_{\pm 0.18}}$ |
| KungFuMaster-v5 | $1.07_{\pm 0.07}$ | $0.74_{\pm 0.08}$ | $1.61_{\pm 0.14}$ | $1.19_{\pm 0.08}$ | $1.01_{\pm 0.02}$ | $\mathbf{2.03_{\pm 0.05}}$ | $1.50_{\pm 0.28}$ | $1.39_{\pm 0.26}$ |
| NameThisGame-v5 | $\mathbf{3.13_{\pm 0.29}}$ | $0.57_{\pm 0.17}$ | $2.00_{\pm 0.23}$ | $0.77_{\pm 0.21}$ | $2.46_{\pm 0.25}$ | $3.17_{\pm 0.13}$ | $2.75_{\pm 0.13}$ | $1.61_{\pm 0.20}$ |
| Phoenix-v5 | $7.99_{\pm 1.21}$ | $0.63_{\pm 0.09}$ | $16.73_{\pm 11.22}$ | $5.42_{\pm 1.31}$ | $3.42_{\pm 2.06}$ | $20.40_{\pm 8.06}$ | $\mathbf{48.14_{\pm 11.77}}$ | $20.24_{\pm 3.78}$ |
| Qbert-v5 | $1.23_{\pm 0.09}$ | $1.32_{\pm 0.06}$ | $1.77_{\pm 0.09}$ | $1.09_{\pm 0.03}$ | $1.08_{\pm 0.08}$ | $1.91_{\pm 0.07}$ | $1.87_{\pm 0.04}$ | $\mathbf{2.84_{\pm 1.24}}$ |
| Riverraid-v5 | $1.23_{\pm 0.19}$ | $0.84_{\pm 0.09}$ | $1.34_{\pm 0.13}$ | $0.91_{\pm 0.14}$ | $0.90_{\pm 0.09}$ | $1.59_{\pm 0.03}$ | $1.57_{\pm 0.25}$ | $\mathbf{1.84_{\pm 0.09}}$ |

Table 6: Performance (mean $\pm$ standard deviation) on the Atari-10 suite over 400M frames, averaged across 3 random seeds.

