# OpenReview forum: "Efficient Multi-Step Reinforcement Learning with Expectation-Maximization Bootstrapping"
_ICLR.cc/2026/Conference — Submitted to ICLR 2026_

### Official Review · Reviewer_hwgN · 2025-10-25

**Soundness:** 3
**Presentation:** 3
**Contribution:** 3
**Rating:** 6
**Confidence:** 3

**Summary:**

The paper proposes Expectation-Maximization Bootstrapping (EMB), a novel framework for multi-step reinforcement learning that interprets the selection of bootstrapping lengths as latent variables.
By introducing a salience–stability measure b(z_n), the method adaptively weighs different n-step returns to balance bias and variance.
Empirical evaluations on Atari benchmarks demonstrate promising performance compared to conventional multi-step methods such as Retrace(λ) and Q(λ).
The paper also provides some theoretical analysis to support the convergence and optimality of the proposed scheme.

**Strengths:**

1. The paper provides intuitive and novel view of bootstrapping.

2. The proposed metric effectively combines the magnitude and consistency of multi-step targets, providing a practical way to balance return quality and stability.

3. The proposed method achieves consistent improvements over baselines on Atari-10 and Atari-57 benchmarks, showing better sample efficiency and robustness.

4. The inclusion of a convergence proposition and connection to return-based Bellman operators adds theoretical credibility.

**Weaknesses:**

1. The definition of b is rather heuristic. More theoretical or empirical discussion would help clarify its role and properties.
For instance, analyzing its effect on variance reduction, learning stability, or numerical behavior under different reward scales would strengthen the paper’s justification.

2. The proposed method is EM-inspired rather than a true EM algorithm.
The E-step is approximated by a heuristic salience–stability score instead of a posterior distribution derived from previous parameters, and the M-step simply re-weights the TD loss without maximizing any explicit likelihood.
However, the paper spends a large portion discussing EM, which may distract readers.
In its current form, the EM framework does not seem to deepen understanding of the actual algorithm or contribute to its improvement.

3. The paper does not specify how b is computed in practice.
For example, what is the exact definition of the normalization terms in Equation (4)?
How is \delta_n calculated (L2 norm per sample or batch average)?
These details are essential for reproducibility.

**Questions:**

1. From the paper, it seems that Figure 1 uses a direct visualization of Eq. (4) rather than the full EMB method.
Could the authors clarify whether this figure illustrates the final adaptive weighting mechanism in EMB or just the salience–stability heuristic itself?

2. The ablation study only explores up to 32-step returns. Since the results suggest that longer horizons yield better performance, why not test larger n (e.g., 64 or 128) to observe potential saturation or instability effects?

---

> ### Author Response · Authors · 2025-11-20
> **Response to Reviewer hwgN (1/2)**
>
> We sincerely appreciate Reviewer hwgN's thoughtful comments. Our detailed responses to your questions and concerns are as follows:
>
> ### Weakness 1: The definition of b is rather heuristic. More theoretical or empirical discussion would help clarify its role and properties. For instance, analyzing its effect on variance reduction, learning stability, or numerical behavior under different reward scales would strengthen the paper’s justification.
> > Answer: The definition of $b(Z\_n)$ is not merely heuristic, but is principled and motivated by established principles from prior RL methods [r3, r4, r5].
> > In this paper, we provide a solid step towards this goal by proposing $b(Z\_n)$ as a novel metric to explicitly quantify the trade-off between salience and stability.
> > First, we provide a strong empirical justification for this design in our motivation experiments. As shown in Figure 1, applying this metric directly can advance Retrace($\lambda$) to gain consistent performance improvements as the maximum bootstrapping step $N$ increases and reduce the performance instability and sensitivity of $n$-step DQN to the choice of $n$.
> > Second, our theoretical analysis (Proposition 4.1) confirms that incorporating $b(Z\_n)$ into the return-based Bellman operator preserves the crucial convergence and optimality guarantees, demonstrating that our novel operator is theoretically sound.
> > While the extensive empirical results in the Atari-57 suite show a solid performance improvement, we agree that more formal theoretical analysis (e.g., variance reduction) is a valuable but challenging research direction. We will explore these theoretical properties in future work.
>
>
> ### Weakness 2: The proposed method is EM-inspired rather than a true EM algorithm. The E-step is approximated by a heuristic salience–stability score instead of a posterior distribution derived from previous parameters, and the M-step simply re-weights the TD loss without maximizing any explicit likelihood. However, the paper spends a large portion discussing EM, which may distract readers. In its current form, the EM framework does not seem to deepen understanding of the actual algorithm or contribute to its improvement.
> > Answer: This paper is motivated by the insight that the selection of the "optimal" bootstrapping step can be viewed as an unobserved latent variable. Thus, the EM formulation serves as the optimal theoretical lens to introduce our "salience and stability" metric into the multi-step RL. Instead of treating this metric as a heuristic, the EM formulation grounds the metric as the estimated posterior derived in the E-step. We provide a detailed explanation of this framework to explicitly frame multi-step RL as a probabilistic inference problem, justifying the alternating structure of our EMB (E-step and M-step).
> > Overall, the EM framework is not a distraction but a fundamental component that deepens the understanding of our method. However, to avoid misunderstanding, we have revised Section 4.2 to explicitly clarify the conceptual nature of this derivation.
>
> ### Weakness 3: The paper does not specify how $b$ is computed in practice. For example, what is the exact definition of the normalization terms in Equation (4)? How is $\delta\_n$ calculated (L2 norm per sample or batch average)? These details are essential for reproducibility.
> > Answer: We have clarified the details of $b(Z\_n)$ in the manuscript. The normalization terms in Eq. (4) are the min-max normlization performed across the set of $N$-step transitions. Specifically, the salience term normalizes the $N$ TD targets $\\{Z_j\\}\_{j=1}^N$, and the stability term normalizes the $N$ corresponding TD residuals $\\{\delta_j\\}\_{j=1}^N$. For $\delta\_j$, we calculate the per-sample absolute difference $|Q(s_t,a_t) - Z\_j|$. This ensures the metric is responsive to the stability of each target.
>
>
> ### Question 1: From the paper, it seems that Figure 1 uses a direct visualization of Eq. (4) rather than the full EMB method. Could the authors clarify whether this figure illustrates the final adaptive weighting mechanism in EMB or just the salience–stability heuristic itself?
> > Answer: Figure 1 serves as our motivation experiment, demonstrating the effectiveness of salience and stability before we formally introduce the full EMB framework in Section 4. Specifically, Figure 1 illustrates two different applications of our core idea:
> > 1. "Retrace($\lambda$) w/ S\&S" illustrates the performance of our EMB($\lambda$), which is formally defined in Section 4.3. We present it early to motivate its consistent performance gains as $N$ increases.
> > 2. "$n$-step DQN w/ S\&S" advances $n$-step DQN by selecting the single bootstrapping step that has the maximum $b(Z\_n)$. This demonstrates the metric's ability to stabilize $n$-step DQN's sensitivity to $n$.
> > We have revised the caption for Figure 1 to make these implementation details explicit and avoid any misunderstanding.

---

> ### Author Response · Authors · 2025-11-20
> **Response to Reviewer hwgN (2/2)**
>
> ### Question 2: The ablation study only explores up to 32-step returns. Since the results suggest that longer horizons yield better performance, why not test larger n (e.g., 64 or 128) to observe potential saturation or instability effects?
> > We adopt $N=32$ to ensure comparability with prior work [r2] and manage the computation efficiency. To address this concern, we have performed additional experiments for $N=64$ and $N=128$. The results (Table R2), which have been added to Section 5.3, show that performance saturates at $N=32$ and shows a slight decline at $N=64$ and $N=128$. These results also indicate that EMB($\lambda$) does not approximate high-variance MC returns, but rather effectively learning from the set of multi-step TD targets.
> >
> > Table R2. Sensitivity of EMB($\lambda$)'s performance to $N$ on the Atari-10 suite, averaged over 3 random seeds.
> >
> > | $N$            | 2   | 4   | 8   | 16  | 32  | 64  | 128 |
> > |--------------|-----|-----|-----|-----|-----|-----|-----|
> > | Atari-10 Score | 305 | 370 | 349 | 483 | $\mathbf{581}$ | 558 | 496 |
>
>
> ## Reference
> > [r1] Munos, Rémi, et al. "Safe and efficient off-policy reinforcement learning." Advances in neural information processing systems 29 (2016).
> >
> > [r2] Gallici, Matteo, et al. "Simplifying Deep Temporal Difference Learning." The Thirteenth International Conference on Learning Representations (2025).
> >
> > [r3] Wang, Yuhui, et al. "Highway reinforcement learning." arXiv preprint arXiv:2405.18289 (2024).
> >
> > [r4] Parisi, Simone, et al. "TD-regularized actor-critic methods." Machine Learning 108.8 (2019): 1467-1501.
> >
> > [r5] Tzannetos, Georgios, et al. "Proximal curriculum for reinforcement learning agents." arXiv preprint arXiv:2304.12877 (2023).

---

> > ### Comment · Reviewer_hwgN · 2025-11-27
> >
> > Thank you for your response. I maintain a positive view of the paper but believe there is room to further improve the organization clarity of the writing.

---

> > > ### Author Response · Authors · 2025-11-27
> > >
> > > We would like to thank the reviewer again for the time and effort dedicated to reviewing our manuscript. We sincerely appreciate the positive assessment of our work and the constructive feedback regarding the writing. In the revised version, we have significantly improved the writing. We believe the structure is now more concise and clear. If there are further suggestions or any other questions, please let us know, and we are happy to address them.

---

### Official Review · Reviewer_f5Nk · 2025-10-25

**Soundness:** 2
**Presentation:** 1
**Contribution:** 2
**Rating:** 2
**Confidence:** 3

**Summary:**

This paper proposes EMB($\lambda$), a method for adaptively selecting the horizon length for n-step return in multi-step RL. The paper aims to theoretically and empirically justify the proposed method, building on previous algorithms and a novel metric for measuring the saliency and stability of a specific multi-step return target.

**Strengths:**

The paper proposes a method for determining parameters for multi-step RL, which is found empirically to improve performance over alternative multi-step RL baselines on the Atari Learning Environment.

The proposed method draws its roots and justification from known and well-used algorithms and methods, thus basing itself well in the existing body of work.

**Weaknesses:**

The introduction is badly written and hard to follow and understand. It is too long, and brings up a multitude of topics and previous work, with barely any unifying line of thought. Much of it is also repeated in the related work section.

Also, it describes papers from 2018-19 as “recent”, which in the fast-moving field of RL is hardly a fair description. What work has been done on the subject of multi-step RL in the past few years? Some recent papers have explored adaptively selecting the mutli-step horizon parameter; it seems like they are missing from the related work in this paper.

The rest of the paper is also unclear and confusing, including many missing definitions for the various bits of notation used. In particular:
- The preliminaries section does not explain what eligibility traces are how they are obtained, or why they help with balancing bias and variance. In particular, $c_j$ is not defined (and is used again later in section 4.3).
- Section 4.1: $N$ is undefined (it is also undefined in the preliminaries, but there it seems like it is the length of some horizon, which does not fully make sense in eq. 4).
- In section 4.3, $\lambda$ is undefined. Is it the standard $\lambda$ used in $TD(\lambda)$ style algorithms? If it isn’t, the lack of definition makes it unclear whether the assumption at the base of the proof for Proposition 4.1 holds.

Regarding experimental results: it is unclear (without variance/standard deviation being presented) whether the gains on baselines are statistically significant.

**Questions:**

1. In line 248, “As shown in Figure 1, we have empirically demonstrated that Equation 4 can efficiently advance existing multi-step RL methods… “: nothing in section 4.1 describes the experiment conducted to produce Figure 1, nor does it describe a full method beyond the definition of a metric. Therefore, how does Equation 4 in itself advance existing multi-step RL methods?

2. Following up on the previous question, in sec. 4.2 the authors describe the $b(Z_n)$ metric as “obtaining optimality” - section 4.1 and Figure 1 do not prove this; at most, they suggest that reward may be correlated with the metric. How do the authors justify this choice of an estimator?

---

> ### Author Response · Authors · 2025-11-20
> **Response to Reviewer f5Nk (1/3)**
>
> We sincerely appreciate Reviewer f5Nk's thoughtful comments. However, we respectfully clarify potential misunderstandings regarding the logic of introduction and currency of the literature, while fully embracing your feedback to enhance the manuscript's clarity. We provide detailed responses to each concern below.
>
> ### Weakness 1: The introduction is badly written and hard to follow and understand. It is too long, and brings up a multitude of topics and previous work, with barely any unifying line of thought. Much of it is also repeated in the related work section.
> > Answer: We sincerely appreciate this comment. Our intention in the introduction was to guide the reader from the limitations of existing multi-step RL methods to the motivation for our proposed "salience and stability" metric, and finally to our EMB framework.
> >
> > While this logical structure was acknowledged by the other reviewers, we fully respect your feedback regarding the length and readability. In response, we have thoroughly improved the introduction to be more concise: we removed redundant background that overlapped with the Related Work section and tightened the narrative so that it focuses solely on motivating the EMB framework. We believe the revised version now presents a much clearer path from the known challenges of multi-step RL to our proposed solution.
>
>
> ### Weakness 2: Also, it describes papers from 2018-19 as “recent”, which in the fast-moving field of RL is hardly a fair description. What work has been done on the subject of multi-step RL in the past few years? Some recent papers have explored adaptively selecting the multi-step horizon parameter; it seems like they are missing from the related work in this paper.
> > Answer: Regarding the currency of our literature review, we clarify that, to the best of our knowledge, we have cited and discussed most state-of-the-art multi-step RL methods. Specifically, Highway-step [r3] is the key baseline that directly addresses the adaptive horizon selection as mentioned by the reviewer. Furthermore, our implementation is built upon PQN [r2], a strong and highly modern base algorithm.
> >
> > Our main contribution complements these works. We introduce EMB, a novel optimization framework to adaptively weight multi-step TD targets based on our new metric for measuring salience and stability (Eq. (4)). We have revised Section 2 to clearly position our contribution and distinction. Finally, we thank the reviewer for pointing out the imprecision in our terminology and have corrected the phrasing regarding "recent" works in the revised manuscript.
>
>
> ### Weakness 3: The rest of the paper is also unclear and confusing, including many missing definitions for the various bits of notation used.
> > Answer: We have performed a careful review of the entire manuscript to ensure all notation is explicitly defined upon its first appearance.
> In particular, we have clarified the following in the revised version:
> > 1. $c\_j$: the trace coefficient for the return-based Bellman operator (lines 193, Section 3).
> > 2. $N$: the maximum bootstrapping step, which defines the full set of $n$-step TD targets, $\\{Z\_n\\}\_{n=1}^N$ (lines 53, Section 1).
> > 3. $\lambda$: standard trace-decay parameter, as used in TD($\lambda$) and Retrace($\lambda$), which controls the weighting of multi-step returns (lines 195, Section 3).

---

> ### Author Response · Authors · 2025-11-20
> **Response to Reviewer f5Nk (2/3)**
>
> ### Weakness 4: Regarding experimental results: it is unclear (without variance/standard deviation being presented) whether the gains on baselines are statistically significant.
> > Answer: While our evaluation metrics follow the same standard practices as in prior work [r2], we agree that providing mean and standard deviation is a clearer way to demonstrate significance.
> > To address this, we have further increased the number of random seeds to 5 for our Atari-10 experiments. We have added Table R2 to report the mean and standard deviation, confirming the statistical significance and robustness of our results, which are added in Appendix H.
> >
> > Table R2. Performance (mean $\pm$ standard deviation) on the Atari-10 suite over 200M frames, averaged across 5 random seeds.
> >
> > | Game            | One-step | $n$-step | Highway-step | IS | TB($\lambda$) | Q($\lambda$) | Retrace($\lambda$) | EMB($\lambda$) (ours) |
> > |-------|----------|--------|-----|----|-------|-------|-----|--------|
> > | Amidar-v5 | $0.37_{\pm 0.09}$ | $1.06_{\pm 0.14}$ | $0.31_{\pm 0.01}$ | $0.52_{\pm 0.14}$ | $0.25_{\pm 0.05}$ | $0.56_{\pm 0.04}$ | $0.53_{\pm 0.15}$ | $\mathbf{0.85_{\pm 0.20}}$ |
> > | BattleZone-v5 | $0.72_{\pm 0.06}$ | $1.06_{\pm 0.14}$ | $0.74_{\pm 0.06}$ | $0.71_{\pm 0.02}$ | $0.80_{\pm 0.06}$ | $\mathbf{1.26_{\pm 0.07}}$ | $1.25_{\pm 0.19}$ | $1.25_{\pm 0.19}$ |
> > | Bowling-v5 | $0.04_{\pm 0.01}$ | $0.06_{\pm 0.01}$ | $0.30_{\pm 0.06}$ | $0.17_{\pm 0.03}$ | $0.04_{\pm 0.02}$ | $0.07_{\pm 0.03}$ | $0.07_{\pm 0.03}$ | $\mathbf{0.37_{\pm 0.09}}$ |
> > | DoubleDunk-v5 | $7.88_{\pm 0.13}$ | $7.92_{\pm 0.39}$ | $10.25_{\pm 1.52}$ | $7.60_{\pm 0.12}$ | $7.71_{\pm 0.28}$ | $7.83_{\pm 0.09}$ | $7.81_{\pm 0.07}$ | $\mathbf{15.51_{\pm 2.67}}$ |
> > | Frostbite-v5 | $0.81_{\pm 0.07}$ | $1.64_{\pm 0.34}$ | $0.85_{\pm 0.08}$ | $1.57_{\pm 0.40}$ | $0.83_{\pm 0.08}$ | $1.34_{\pm 0.33}$ | $1.42_{\pm 0.17}$ | $\mathbf{1.93_{\pm 0.32}}$ |
> > | KungFuMaster-v5 | $1.07_{\pm 0.17}$ | $1.44_{\pm 0.07}$ | $1.09_{\pm 0.14}$ | $0.66_{\pm 0.08}$ | $1.15_{\pm 0.20}$ | $\mathbf{1.80_{\pm 0.35}}$ | $1.74_{\pm 0.25}$ | $1.53_{\pm 0.21}$ |
> > | NameThisGame-v5 | $1.58_{\pm 0.38}$ | $1.67_{\pm 0.31}$ | $1.17_{\pm 0.15}$ | $0.34_{\pm 0.08}$ | $1.79_{\pm 0.19}$ | $2.35_{\pm 0.31}$ | $\mathbf{2.47_{\pm 0.28}}$ | $1.34_{\pm 0.18}$ |
> > | Phoenix-v5 | $3.26_{\pm 1.38}$ | $5.93_{\pm 8.36}$ | $3.58_{\pm 0.49}$ | $0.66_{\pm 0.01}$ | $5.91_{\pm 2.24}$ | $12.43_{\pm 9.51}$ | $15.86_{\pm 3.09}$ | $\mathbf{23.70_{\pm 8.67}}$ |
> > | Qbert-v5 | $1.01_{\pm 0.06}$ | $1.46_{\pm 0.14}$ | $0.95_{\pm 0.07}$ | $1.09_{\pm 0.04}$ | $1.08_{\pm 0.08}$ | $1.51_{\pm 0.14}$ | $1.62_{\pm 0.15}$ | $\mathbf{1.76_{\pm 0.06}}$ |
> > | Riverraid-v5 | $0.86_{\pm 0.15}$ | $1.34_{\pm 0.17}$ | $0.86_{\pm 0.09}$ | $0.71_{\pm 0.04}$ | $0.89_{\pm 0.21}$ | $1.38_{\pm 0.15}$ | $\mathbf{1.44_{\pm 0.10}}$ | $1.41_{\pm 0.10}$ |
>
>
> ### Question 1: In line 248, “As shown in Figure 1, we have empirically demonstrated that Equation 4 can efficiently advance existing multi-step RL methods… “: nothing in section 4.1 describes the experiment conducted to produce Figure 1, nor does it describe a full method beyond the definition of a metric. Therefore, how does Equation 4 in itself advance existing multi-step RL methods?
> > Answer: Figure 1 serves as our motivation experiment, demonstrating the effectiveness of our salience and stability technique before we formally introduce the full EMB framework in Section 4. Specifically, we applied this novel metric in existing multi-step RL methods, including $n$-step DQN and Retrace($\lambda$).
> > 1. "Retrace($\lambda$) w/ S\&S" illustrate the performance of our EMB($\lambda$), which is formally defined in Section 4.3. We present it early to motivate its consistent performance gains as the maximum bootstrapping step $N$ increases.
> > 2. "$n$-step DQN w/ S\&S" advances $n$-step DQN by selecting the single bootstrapping step that has the maximum $b(Z\_n)$. This demonstrates the metric's ability to stabilize $n$-step DQN's sensitivity to $n$.
> > We have revised the caption and added Appendix F to include implementation details to avoid any misunderstanding.
>
>
> ### Question 2: Following up on the previous question, in sec. 4.2 the authors describe the $b(Z\_n)$ metric as “obtaining optimality” - section 4.1 and Figure 1 do not prove this; at most, they suggest that reward may be correlated with the metric. How do the authors justify this choice of an estimator?
> > Answer: In our EMB framework, we frame the problem as maximizing the log-evidence of an "optimality" event $O$. Specifically, $p(O|Z\_n, Q)$ represents the likelihood that $Z\_n$ is a salient and stable choice for bootstrapping, which guides the Q-function towards the optimal Q-function. In practice, we model this unobserved likelihood using $b(Z\_n)$ as an empirical proxy for measuring salience and stability. We have clarified this terminology in Section 4.2.

---

> ### Author Response · Authors · 2025-11-20
> **Response to Reviewer f5Nk (3/3)**
>
> ### Reference
> > [r1] Munos, Rémi, et al. "Safe and efficient off-policy reinforcement learning." Advances in neural information processing systems 29 (2016).
> >
> > [r2] Gallici, Matteo, et al. "Simplifying Deep Temporal Difference Learning." The Thirteenth International Conference on Learning Representations (2025).
> >
> > [r3] Wang, Yuhui, et al. "Highway reinforcement learning." arXiv preprint arXiv:2405.18289 (2024).

---

### Official Review · Reviewer_Q13V · 2025-10-31

**Soundness:** 3
**Presentation:** 3
**Contribution:** 3
**Rating:** 6
**Confidence:** 4

**Summary:**

The paper proposes a new off-policy online reinforcement learning algorithm called Expectation-Maximization Bootstrapping (EMB). The paper addresses the bias-variance trade-off when learning with multi-step TD targets. EMB learns a metric to evaluate the salience and stability of each bootstrapping step, and uses the measure to weight the TD targets. The derived operator remains a contraction, ensuring the convergence of value estimation under a fixed policy. Empirically, EMB outperforms baseline methods across a wide range of environments.

The key contributions are: (1) introducing a metric for quantifying the balance between salience and stability in equation 4; (2) proving EMB’s convergence and optimality in Proposition 4.1; and (3) demonstrating EMB’s advantage regarding both the efficiency and final policy under a fixed learning budget, compared to baseline methods, across multiple atari games (Figure 2).

**Strengths:**

1. The method is novel, integrating the expectation–maximization framework into reinforcement learning to dynamically balance salience and stability in multi-step learning. In existing literature, n-step and $\lambda$-return methods rely on a fixed n or $\lambda$ to trade off salience and stability, which limits their adaptability. EMB lifts this limitation by introducing dynamic weighting over multi-step temporal-difference targets, adaptively emphasizing those that are more salient.

2. The method is well supported by theoretical results. Section 4.3 demonstrates that the derived operator is a contraction, which guarantees convergence and optimality of value estimation under a fixed policy.

3. The empirical evaluation is extensive, covering over 60 Atari environments (10 on Atari-10 and the rest on Atari-57), providing convincing evidence of EMB’s learning efficiency.

4. This work performs an ablation study on key parameters ($\lambda$ and $N$), empirically indicating how these choices affect learning efficiency.

**Weaknesses:**

1. There is remaining ambiguity regarding the optimal choice of N. Figure 3(d) shows that larger N improves performance. Equation (4) indicates that the boundaries of normalization of TD targets and TD residuals depend on N, which suggests that a smaller N may lead to an inaccurate rating due to an overly small range for selecting the minimum and maximum values. This raises the question of whether EMB’s performance gain is mainly due to approaching Monte Carlo-style learning. A more detailed discussion comparing EMB to MC learning would strengthen the paper. Adding an MC-style algorithm into the experimental baseline could also be beneficial.

2. While the evaluation spans many environments, results are averaged over only 3 seeds, which is not sufficient to compute reliable confidence intervals.

3. The learning curves are cut off before showing the sign of convergence, for example, Phoenix in Figure 2, Alien, Asterix, Berzerk, Bowling, and RoadRunner in Figure 4. I would not say it is a significant weakness because baselines showed lower learning efficiency in these environments, however, it may still be worth checking longer runs on these environments to see how EMB converges to gain information on its stability.

**Questions:**

In the learning curves, does the shaded area represent standard deviation, standard error, or minimum/maximum performance across all runs?

---

> ### Author Response · Authors · 2025-11-20
> **Response to Reviewer Q13V (1/2)**
>
> We sincerely appreciate Reviewer Q13V's thoughtful comments. Below, we give responses to your questions and concerns.
>
> ### Weakness 1: There is remaining ambiguity regarding the optimal choice of N. Figure 3(d) shows that larger N improves performance. Equation (4) indicates that the boundaries of normalization of TD targets and TD residuals depend on N, which suggests that a smaller N may lead to an inaccurate rating due to an overly small range for selecting the minimum and maximum values. This raises the question of whether EMB’s performance gain is mainly due to approaching Monte Carlo-style learning. A more detailed discussion comparing EMB to MC learning would strengthen the paper. Adding an MC-style algorithm into the experimental baseline could also be beneficial.
> > Answer: While increasing $N$ expands the normalization range in Eq. (4), $b(Z\_n)$ is adaptively normalized, ensuring that weighting remains stable across different $N$. To address your concern, we performed additional experiments with larger $N$ ($N=64$ and $N=128$) on the Atari-10 suite. The results (Table R2) show that the performance saturates at $N=32$ and slightly declines at $N=64$ and $N=128$. These results indicate that EMB($\lambda$) is not simply approximating high variance MC returns. Unlike pure MC-style methods, which suffer from the high variance due to the accumulation of stochastic dynamics and rewards, EMB($\lambda$) mitigates this issue by explicitly retaining bootstrapping through salient and stable updates. We have clarified this distinction and these discussions in Section 5.3.
> >
> > Table R2. Sensitivity of EMB($\lambda$)'s performance to $N$ on the Atari-10 suite, averaged over 3 random seeds.
> >
> > | $N$            | 2   | 4   | 8   | 16  | 32  | 64  | 128 |
> > |--------------|-----|-----|-----|-----|-----|-----|-----|
> > | Atari-10 Score | 305 | 370 | 349 | 483 | $\mathbf{581}$ | 558 | 496 |
>
>
> ### Weakness 2: While the evaluation spans many environments, results are averaged over only 3 seeds, which is not sufficient to compute reliable confidence intervals.
> > Answer: We initially adopted 3 random seeds, as this is a common protocol for the computationally intensive Atari Learning Environment benchmark, consistent with many existing works [r2].
> >
> > However, we agree that more seeds improve reliability. To address this concern and further validate our method's stability, we have run additional experiments with 5 random seeds on the Atari-10 suite. As shown in the Table, the results are consistent with our original findings and confirm the performance robustness of EMB($\lambda$).  We have included this clarification and additional results in the revised version (Appendix H).
> >
> > Table R2. Performance (mean $\pm$ standard deviation) on the Atari-10 suite over 200M frames, averaged across 5 random seeds.
> >
> > | Game            | One-step | $n$-step | Highway-step | IS | TB($\lambda$) | Q($\lambda$) | Retrace($\lambda$) | EMB($\lambda$) (ours) |
> > |-------|-----|-----|-----|----|----|--------|---------|------------------|
> > | Amidar-v5 | $0.37_{\pm 0.09}$ | $1.06_{\pm 0.14}$ | $0.31_{\pm 0.01}$ | $0.52_{\pm 0.14}$ | $0.25_{\pm 0.05}$ | $0.56_{\pm 0.04}$ | $0.53_{\pm 0.15}$ | $\mathbf{0.85_{\pm 0.20}}$ |
> > | BattleZone-v5 | $0.72_{\pm 0.06}$ | $1.06_{\pm 0.14}$ | $0.74_{\pm 0.06}$ | $0.71_{\pm 0.02}$ | $0.80_{\pm 0.06}$ | $\mathbf{1.26_{\pm 0.07}}$ | $1.25_{\pm 0.19}$ | $1.25_{\pm 0.19}$ |
> > | Bowling-v5 | $0.04_{\pm 0.01}$ | $0.06_{\pm 0.01}$ | $0.30_{\pm 0.06}$ | $0.17_{\pm 0.03}$ | $0.04_{\pm 0.02}$ | $0.07_{\pm 0.03}$ | $0.07_{\pm 0.03}$ | $\mathbf{0.37_{\pm 0.09}}$ |
> > | DoubleDunk-v5 | $7.88_{\pm 0.13}$ | $7.92_{\pm 0.39}$ | $10.25_{\pm 1.52}$ | $7.60_{\pm 0.12}$ | $7.71_{\pm 0.28}$ | $7.83_{\pm 0.09}$ | $7.81_{\pm 0.07}$ | $\mathbf{15.51_{\pm 2.67}}$ |
> > | Frostbite-v5 | $0.81_{\pm 0.07}$ | $1.64_{\pm 0.34}$ | $0.85_{\pm 0.08}$ | $1.57_{\pm 0.40}$ | $0.83_{\pm 0.08}$ | $1.34_{\pm 0.33}$ | $1.42_{\pm 0.17}$ | $\mathbf{1.93_{\pm 0.32}}$ |
> > | KungFuMaster-v5 | $1.07_{\pm 0.17}$ | $1.44_{\pm 0.07}$ | $1.09_{\pm 0.14}$ | $0.66_{\pm 0.08}$ | $1.15_{\pm 0.20}$ | $\mathbf{1.80_{\pm 0.35}}$ | $1.74_{\pm 0.25}$ | $1.53_{\pm 0.21}$ |
> > | NameThisGame-v5 | $1.58_{\pm 0.38}$ | $1.67_{\pm 0.31}$ | $1.17_{\pm 0.15}$ | $0.34_{\pm 0.08}$ | $1.79_{\pm 0.19}$ | $2.35_{\pm 0.31}$ | $\mathbf{2.47_{\pm 0.28}}$ | $1.34_{\pm 0.18}$ |
> > | Phoenix-v5 | $3.26_{\pm 1.38}$ | $5.93_{\pm 8.36}$ | $3.58_{\pm 0.49}$ | $0.66_{\pm 0.01}$ | $5.91_{\pm 2.24}$ | $12.43_{\pm 9.51}$ | $15.86_{\pm 3.09}$ | $\mathbf{23.70_{\pm 8.67}}$ |
> > | Qbert-v5 | $1.01_{\pm 0.06}$ | $1.46_{\pm 0.14}$ | $0.95_{\pm 0.07}$ | $1.09_{\pm 0.04}$ | $1.08_{\pm 0.08}$ | $1.51_{\pm 0.14}$ | $1.62_{\pm 0.15}$ | $\mathbf{1.76_{\pm 0.06}}$ |
> > | Riverraid-v5 | $0.86_{\pm 0.15}$ | $1.34_{\pm 0.17}$ | $0.86_{\pm 0.09}$ | $0.71_{\pm 0.04}$ | $0.89_{\pm 0.21}$ | $1.38_{\pm 0.15}$ | $\mathbf{1.44_{\pm 0.10}}$ | $1.41_{\pm 0.10}$ |

---

> ### Author Response · Authors · 2025-11-20
> **Response to Reviewer Q13V (2/2)**
>
> ### Weakness 3: The learning curves are cut off before showing the sign of convergence, for example, Phoenix in Figure 2, Alien, Asterix, Berzerk, Bowling, and RoadRunner in Figure 4. I would not say it is a significant weakness because baselines showed lower learning efficiency in these environments, however, it may still be worth checking longer runs on these environments to see how EMB converges to gain information on its stability.
> > Answer: Our initial setting of 200M frames was adopted to ensure a standard and fair comparison with prior multi-step RL baselines, as this is the common setting in the literature [r2]. To address this concern, we have extended the runs on the Atari-10 suite from 200M to 400M frames. The results (Table R3) show that EMB($\lambda$) maintains its stable performance lead, confirming its long-term stability, as our previous findings. We have added this new analysis to Appendix H.
> >
> > Table R3. Performance (mean $\pm$ standard deviation) on the Atari-10 suite over 400M frames, averaged across 3 random seeds.
> >
> > | Game            | One-step | $n$-step | Highway-step | IS | TB($\lambda$) | Q($\lambda$) | Retrace($\lambda$) | EMB($\lambda$) |
> > |-----------------|----------|--------|--------------|----|----------------|---------------|-------------------|----------------|
> > | Amidar-v5 | $0.66_{\pm 0.14}$ | $0.63_{\pm 0.05}$ | $0.63_{\pm 0.13}$ | $0.40_{\pm 0.02}$ | $0.39_{\pm 0.11}$ | $0.73_{\pm 0.03}$ | $0.78_{\pm 0.11}$ | $\mathbf{1.37_{\pm 0.16}}$ |
> > | BattleZone-v5 | $0.91_{\pm 0.07}$ | $0.87_{\pm 0.07}$ | $1.10_{\pm 0.02}$ | $0.73_{\pm 0.08}$ | $1.00_{\pm 0.22}$ | $1.50_{\pm 0.11}$ | $1.25_{\pm 0.11}$ | $\mathbf{1.53_{\pm 0.06}}$ |
> > | Bowling-v5 | $0.06_{\pm 0.02}$ | $0.26_{\pm 0.05}$ | $0.03_{\pm 0.01}$ | $0.27_{\pm 0.12}$ | $0.12_{\pm 0.10}$ | $0.04_{\pm 0.04}$ | $0.06_{\pm 0.00}$ | $\mathbf{0.32_{\pm 0.05}}$ |
> > | DoubleDunk-v5 | $8.00_{\pm 0.00}$ | $7.43_{\pm 0.25}$ | $7.77_{\pm 0.25}$ | $12.32_{\pm 5.32}$ | $7.62_{\pm 0.21}$ | $7.89_{\pm 0.25}$ | $7.92_{\pm 0.19}$ | $\mathbf{14.67_{\pm 5.20}}$ |
> > | Frostbite-v5 | $0.84_{\pm 0.07}$ | $2.13_{\pm 0.48}$ | $1.54_{\pm 0.44}$ | $1.17_{\pm 0.19}$ | $0.80_{\pm 0.05}$ | $1.54_{\pm 0.34}$ | $1.65_{\pm 0.53}$ | $\mathbf{2.29_{\pm 0.18}}$ |
> > | KungFuMaster-v5 | $1.07_{\pm 0.07}$ | $0.74_{\pm 0.08}$ | $1.61_{\pm 0.14}$ | $1.19_{\pm 0.08}$ | $1.01_{\pm 0.02}$ | $\mathbf{2.03_{\pm 0.05}}$ | $1.50_{\pm 0.28}$ | $1.39_{\pm 0.26}$ |
> > | NameThisGame-v5 | $\mathbf{3.13_{\pm 0.29}}$ | $0.57_{\pm 0.17}$ | $2.00_{\pm 0.23}$ | $0.77_{\pm 0.21}$ | $2.46_{\pm 0.25}$ | $3.17_{\pm 0.13}$ | $2.75_{\pm 0.13}$ | $1.61_{\pm 0.20}$ |
> > | Phoenix-v5 | $7.99_{\pm 1.21}$ | $0.63_{\pm 0.09}$ | $16.73_{\pm 11.22}$ | $5.42_{\pm 1.31}$ | $3.42_{\pm 2.06}$ | $20.40_{\pm 8.06}$ | $\mathbf{48.14_{\pm 11.77}}$ | $20.24_{\pm 3.78}$ |
> > | Qbert-v5 | $1.23_{\pm 0.09}$ | $1.32_{\pm 0.06}$ | $1.77_{\pm 0.09}$ | $1.09_{\pm 0.03}$ | $1.08_{\pm 0.08}$ | $1.91_{\pm 0.07}$ | $1.87_{\pm 0.04}$ | $\mathbf{2.84_{\pm 1.24}}$ |
> > | Riverraid-v5 | $1.23_{\pm 0.19}$ | $0.84_{\pm 0.09}$ | $1.34_{\pm 0.13}$ | $0.91_{\pm 0.14}$ | $0.90_{\pm 0.09}$ | $1.59_{\pm 0.03}$ | $1.57_{\pm 0.25}$ | $\mathbf{1.84_{\pm 0.09}}$ |
>
>
> ### Question 1: In the learning curves, does the shaded area represent standard deviation, standard error, or minimum/maximum performance across all runs?
> > Answer: The shaded areas represent the minimum and maximum performance across three independent runs. Furthermore, to provide a more robust statistical measure, we have also added tables (Table R2 and Table R3) reporting the mean and standard deviation for our main results in Appendix H and Appendix I.
>
> ### Reference
> > [r1] Munos, Rémi, et al. "Safe and efficient off-policy reinforcement learning." Advances in neural information processing systems 29 (2016).
> >
> > [r2] Gallici, Matteo, et al. "Simplifying Deep Temporal Difference Learning." The Thirteenth International Conference on Learning Representations (2025).
> >
> > [r3] Wang, Yuhui, et al. "Highway reinforcement learning." arXiv preprint arXiv:2405.18289 (2024).

---

### Official Review · Reviewer_ipZi · 2025-11-01

**Soundness:** 2
**Presentation:** 3
**Contribution:** 3
**Rating:** 4
**Confidence:** 4

**Summary:**

This paper proposes Expectation–Maximization Bootstrapping (EMB), a framework for efficient multi-step off-policy reinforcement learning.
The central idea is to treat the bootstrapping horizon $Z_n$ as a latent variable and optimize it using an Expectation–Maximization (EM) procedure: in the E-step, a salience-and-stability weighting function $b(Z_n)$ is computed to balance bias and variance, and in the M-step, this weighting redefines the multi-step target used for temporal-difference updates.

**Strengths:**

1. Treating the bootstrapping horizon as a latent variable optimized through EM is conceptually elegant and provides a probabilistic interpretation of multi-step bootstrapping.
The introduction of $b(Z_n)$ as a salience–stability weighting is intuitive and aligns with the bias–variance tradeoff in temporal-difference learning.

2. The empirical study covers both Atari-10 and Atari-57 benchmarks with meaningful metrics (IQM, median, and human-normalized scores).
EMB demonstrates consistent sample-efficiency gains and robust performance on many games.

3. The narrative connecting bias–variance control, multi-step targets, and EM optimization is cohesive and well-motivated.
Figures 2–3 are clean and effectively illustrate the intended performance improvements.

**Weaknesses:**

1. All formal lemmas (Lemmas 4.1 and 4.2, Proposition 4.1) are restatements of the contraction and optimality theorems from Munos et al. (2016). EMB merely substitutes its weighting $b(Z_n)$ into the Retrace operator and claims inheritance of those properties.
2. The paper never lists the assumptions under which the Munos-style contraction proof holds—finite state/action spaces, bounded rewards, bounded importance ratios, stationary policies, and tabular expectation updates.
Consequently, the claimed convergence of $T_{\text{EMB}}$ applies only to the tabular setting, not to the deep-RL experiments.
The omission risks confusing readers about the theoretical scope.
3. In the Preliminaries, the symbol $\Delta(\cdot)$ appears in the probability expression for latent variables but is never defined.
It likely denotes a Dirac delta distribution or deterministic event; this should be explicitly clarified.
4. From Figure 2, EMB performs worse than Retrace on KungFuMaster and NameThisGame, two dense-reward, short-horizon environments. These underperformances suggest that $b(Z_n)$ may overemphasize long-horizon updates or that the method is sensitive to global hyperparameters. The paper would benefit from discussing this behavior or offer per-game analysis.

**Questions:**

1. Could the authors explicitly restate the assumptions under which the convergence of $T_{\text{EMB}}$ holds? Is the contraction proof valid only in the tabular setting?

2. Have the authors tested per-game tuning of $\lambda$ or the step limit $N$? Could such tuning mitigate the performance drops on KungFuMaster and NameThisGame?

3. How sensitive is EMB to the computation or normalization of $b(Z_n)$ across games? Does it add variance during learning?

---

> ### Author Response · Authors · 2025-11-20
> **Response to Reviewer ipZi (1/2)**
>
> We sincerely appreciate Reviewer ipZi for thoughtful comments. Below, we give responses to your questions and concerns.
>
> ### Weakness 1: All formal lemmas (Lemmas 4.1 and 4.2, Proposition 4.1) are restatements of the contraction and optimality theorems from Munos et al. (2016). EMB merely substitutes its weighting $b(Z\_n)$ into the Retrace operator and claims inheritance of those properties.
> > Answer: Our main focus in this paper is to propose a novel, efficient, and theoretically reasonable multi-step RL method. Specifically, we introduce the EMB return-based Bellman operator, which incorporates adaptive multi-step weights $b(Z_n)$ and yields strong empirical improvements over fixed-trace methods such as Retrace($\lambda$) [r1].
> >
> > The theoretical results aim to demonstrate that our method is theoretically well-founded. They are not intended as novel theoretical contributions, but rather serve as a soundness proof showing that replacing fixed traces with adaptive weights preserves the key contraction and optimality guarantees. Specifically, Proposition 4.1 confirms that EMB($\lambda$) remains theoretically reasonable and safe to use.
>
>
> ### Weakness 2: The paper never lists the assumptions under which the Munos-style contraction proof holds—finite state/action spaces, bounded rewards, bounded importance ratios, stationary policies, and tabular expectation updates. Consequently, the claimed convergence of $\mathcal{T}\_\text{EMB}$ applies only to the tabular setting, not to the deep-RL experiments. The omission risks confusing readers about the theoretical scope.
> > Answer: We thank the reviewer for this important point. This paper considers the same assumptions as Retrace($\lambda$), including finite state and action spaces, bounded rewards and tabular expectation updates.
> >
> > Since our theoretical analysis of $\mathcal{T}\_\text{EMB}$ is built upon Retrace($\lambda$) [r1], it necessarily relies on the same assumptions. We believe these shared assumptions help position our work as a neat improvement against the Retrace($\lambda$), demonstrating that we can introduce adaptive weights related to salience and stability while provably maintaining the same theoretical convergence guarantees in the tabular case.
> >
> > We agree with the reviewer that an explicit clarification of these general assumptions is crucial for readability and for correctly scoping the theoretical claims. We have revised Sections 3 and 4.3 to state these assumptions explicitly. Furthermore, we have added a clear statement in Section 4.3 to emphasize that our deep RL experiments serve as an empirical demonstration of EMB($\lambda$)'s practical effectiveness, distinct from the formal tabular convergence proof.
>
>
> ### Weakness 3: In the Preliminaries, the symbol $\Delta(\cdot)$ appears in the probability expression for latent variables but is never defined. It likely denotes a Dirac delta distribution or deterministic event; this should be explicitly clarified.
> > Answer: $\Delta(\mathcal{S})$ denotes the set of probability distributions over the state space $\mathcal{S}$.
> We have revised the notation in Section 3 to explicitly define this definition and ensure all other notations are clearly defined.

---

> ### Author Response · Authors · 2025-11-20
> **Response to Reviewer ipZi (2/2)**
>
> ### Weakness 4: From Figure 2, EMB performs worse than Retrace on KungFuMaster and NameThisGame, two dense-reward, short-horizon environments. These underperformances suggest that $b(Z\_n)$ may overemphasize long-horizon updates or that the method is sensitive to global hyperparameters. The paper would benefit from discussing this behavior or offer per-game analysis.
> > Answer: In this paper, we mainly focus on the average performance across the Atari suite, which is the standard setting in the literature [r2] to ensure fair comparison.
> >
> > However, we agree that per-game analysis is meaningful to understand the agent's behaviour in specific scenarios like KungFuMaster and NameThisGame. Therefore, we conducted additional per-game tuning experiments on these two games. The results shown in Table R1 indicate that reducing the maximum bootstrapping steps, $N$, significantly improves the performance of EMB($\lambda$). This confirms that the underperformance was not due to instability, but rather because the global hyperparameter setting allowed $b(Z\_n)$ to prioritize long-horizon updates in environments where shorter multi-step returns are more effective.
> >
> > Table R1. Sensitivity of EMB($\lambda$)'s performance to $N$ on KungFuMaster-v5 and NameThisGame-v5.
> >
> > | $N$            | 2               | 4               | 8               | 16              | 32              | 64              | 128             |
> > |--------------|-----------------|----------------|----------------|----------------|----------------|----------------|----------------|
> > | KungFuMaster-v5  | $1.63_{\pm 0.13}$    | $\mathbf{1.73_{\pm 0.02}}$ | $1.37_{\pm 0.25}$    | $1.44_{\pm 0.16}$    | $1.69_{\pm 0.08}$    | $1.50_{\pm 0.08}$    | $1.42_{\pm 0.12}$    |
> > | NameThisGame-v5  | $\mathbf{2.41_{\pm 0.06}}$ | $2.01_{\pm 0.11}$    | $1.92_{\pm 0.09}$    | $1.50_{\pm 0.19}$    | $1.35_{\pm 0.18}$    | $1.12_{\pm 0.15}$    | $1.23_{\pm 0.12}$    |
> >
> > While this resolves the concern for these specific games, we acknowledge that more analysis is needed to enable the EMB($\lambda$) to automatically adapt $N$. We plan to investigate this adaptive mechanism in future work and have included a discussion of these findings in Appendix G.
>
>
> ### Question 1: Could the authors explicitly restate the assumptions under which the convergence of $T\_{EMB}$ holds? Is the contraction proof valid only in the tabular setting?
> > Answer: As detailed in our response to Weakness 2, we have now explicitly stated that the contraction and optimality results follow the same assumptions as Retrace($\lambda$) in Sections 3 and 4.3. Consequently, the formal convergence proof applies only in the tabular setting. We have also clarified that our deep RL experiments are an empirical demonstration of the practical effectiveness.
>
>
> ### Question 2: Have the authors tested per-game tuning of $\lambda$ or the step limit $N$? Could such tuning mitigate the performance drops on KungFuMaster and NameThisGame?
> > Answer: As detailed in our response to Weakness 4, we performed additional per-game tuning of $N$ for KungFuMaster and NameThisGame. The results confirm the reviewer’s hypothesis that reducing the $N$ yields improved performance in these two games. This indicates that the observed performance degeneration reflects environment-specific preferences rather than inherent sensitivity or instability of EMB($\lambda$). We have included this additional discussion in Appendix G.
>
>
> ### Question 3: How sensitive is EMB to the computation or normalization of $b(Z\_n)$ across games? Does it add variance during learning?
> > Answer: EMB($\lambda$) exhibits low sensitivity to the computation of $b(Z\_n)$ across different games because of the normalization operation. Specifically, the max-min normalization applied in Eq.(4) constrains $b(Z\_n)\in[0, 1]$, ensuring the consistent weighting scale across all games, regardless of the reward structures or Q-value magnitudes. Additionally, rather than adding variance, $b(Z\_n)$ is explicitly designed to reduce variance by down-weighting bootstrapping steps with large TD residuals, which is the primary source of learning instability. EMB($\lambda$) aims to find salient and stable updates, thereby improving the overall stability of the learning process.
>
>
> ### Reference
> > [r1] Munos, Rémi, et al. "Safe and efficient off-policy reinforcement learning." Advances in neural information processing systems 29 (2016).
> >
> > [r2] Gallici, Matteo, et al. "Simplifying Deep Temporal Difference Learning." The Thirteenth International Conference on Learning Representations (2025).
> >
> > [r3] Wang, Yuhui, et al. "Highway reinforcement learning." arXiv preprint arXiv:2405.18289 (2024).

---

> > ### Comment · Reviewer_ipZi · 2025-11-28
> >
> > I appreciate the authors for their time and effort. The revision addresses several of my earlier concerns. The assumptions needed for the convergence argument are now stated clearly, and the revision makes explicit that the theoretical guarantees apply only in the tabular setting inherited from Retrace. The added per-game analysis and the sensitivity study on the choice of $N$ provide a clear explanation for the earlier underperformance on KungFuMaster and NameThisGame.
> >
> > Two points remain. First, the behavior of the weighting term $b(Z_n)$ still depends heavily on the normalization over the full $N$-step window, and large variation in ${Z_j}$ or ${\delta_j}$ can amplify variance in practice; this effect is not analyzed. Second, the theory does not extend beyond the tabular case, even though all experiments use deep function approximation.
> >
> > Overall, the revisions improve the clarity and empirical support of the work, so I am raising my score to 6. I still believe the work is somewhat incremental in its theoretical contribution, but the empirical results and clarifications make it reasonable for a marginal acceptance.

---

> > > ### Author Response · Authors · 2025-12-02
> > >
> > > We appreciate the reviewer's acknowledgement that our revisions improved the clarity and empirical support, and we are grateful for the score increase to 6. We also thank the reviewer for the time and effort dedicated to evaluating our manuscript and for the constructive feedback. Below, we address the reviewer’s follow-up questions.
> > > - **For the sensitivity and variance of the $b(Z\_n)$**: While the normalization in Eq. (4)  could be sensitive to outliers, it is computed locally within the specific $N$-step trajectory of the current update rather than globally over the entire horizon. This acts as an adaptive normalizer, allowing the EMB($\lambda$) to handle different reward scales effectively with stable performance and low variance in practice, as demonstrated by our experimental results.
> > > - **For the concern of theoretical analysis for function approximation**: We conducted extensive empirical validation on the Atari benchmark to demonstrate efficacy in the function approximation setting. In Section 4.3, we have clarified that while our proof establishes the soundness of the operator in the tabular case, the function approximation setting is validated empirically. Theoretical analysis of non-linear function approximation remains challenging and is beyond the scope of this work. We have clarified the scope of our theoretical analysis in Section 4.3 and listed the investigation of function approximation bounds as a direction for future work.

---

### Author Response · Authors · 2025-12-02
**Summary for AC on Review and Rebuttal**

We sincerely thank all the reviewers for their constructive feedback, which has significantly strengthened our paper. We are delighted that the reviewers acknowledged the novelty and contribution of our work, specifically highlighting the **novelty** of bootstrapping in multi-step reinforcement learning from the perspective of Expectation-Maximization and the **significant performance improvements** over baselines across various Atari games.

We have comprehensively responded to all questions and addressed the concerns regarding theoretical scope, empirical robustness, and presentation clarity. These efforts were explicitly acknowledged by the active reviewers, resulting in **three reviewers holding positive evaluations**. Specifically,
- Reviewer ipZi raised the score to 6 following our rebuttal, acknowledging that the assumptions are now stated clearly and that the additional analysis provides a clear explanation for performance.
- Reviewers Q13V and hwgN both maintain positive evaluations (score 6), acknowledging our clarification of the algorithm and improved empirical rigour.
- Due to the known issue, reviewer f5Nk has not and cannot respond within the discussion period. We note that this reviewer primarily focused on writing and showed some misunderstandings regarding the literature and motivation. We fully respected the provided suggestions and have thoroughly revised the manuscript to improve clarity and conciseness.

We have addressed all concerns as follows:

**Clarification of Theoretical Scope**:
- Assumptions (Reviewer ipZi): We clarified that our analysis adopts standard tabular assumptions (e.g., finite state and action spaces) consistent with Retrace($\lambda$). We explicitly revised Sections 3 and 4.3 to state this scope.
- Theoretical scope (Reviewer ipZi): We emphasized that Proposition 4.1 serves as a soundness proof for our proposed operator in the tabular setting, distinct from the deep RL empirical results. Reviewer ipZi confirmed that these assumptions are now "stated clearly" and that the revision correctly scopes the theoretical guarantees.

**Extensive Empirical Results**:
- Statistical significance (Reviewers Q13V and f5Nk): We provided additional results with 5 random seeds, confirming the robustness of EMB($\lambda$).
- Convergence and stability (Reviewer Q13V): We extended training runs from 200M to 400M frames, demonstrating that EMB($\lambda$) maintains long-term stability without collapse.
- Monte Carlo approximation (Reviewers Q13V and hwgN): We conducted sensitivity analysis on larger maximum bootstrapping steps $N$, confirming that EMB($\lambda$) gains stem from adaptive (salience and stability) weighting rather than simply approximating high-variance Monte Carlo returns.
- Specific games analysis (Reviewer ipZi): We performed additional tuning and analysis on specific games (e.g., KungFuMaster and NameThisGame), validating that EMB($\lambda$) is stable but responsive to environment-specific horizons.

**Presentation and Notation**:
- Writing (Reviewer f5Nk): We significantly improved the manuscript by rewriting the introduction to be focused and concise, removing redundant background.
- Motivation Experiments (Reviewers f5Nk and hwgN): We added comprehensive implementation details for the motivation experiments (Fig. 1) to clarify the distinction between the metrics' direct application and the full EMB framework.
- EM framework (Reviewer hwgN): We clarified the role of the EM framework, moving beyond heuristic interpretations to explicitly ground our salience and stability metric as an estimated probabilistic posterior.
- Notation (Reviewers ipZi and f5Nk): We added missing definitions (e.g., $\Delta(\cdot)$, $c_j$) as requested by Reviewers ipZi and f5Nk to ensure mathematical precision.

We thank the Area Chair for their time and dedication in evaluating our manuscript. We hope the Area Chair will consider our comprehensive rebuttal and the resulting positive consensus among the active reviewers in their final decision.

---

### Meta-Review · Area_Chair_bgp3 · 2026-01-06

**Summary:**

**Paper Summary**

This paper introduces Expectation-Maximization Bootstrapping (EMB), a framework that treats the bootstrapping step $n$ as a latent variable, to address the bias-variance tradeoff in multi-step RL. By employing a novel 'Salience and Stability' (S&S) metric within an EM inference process, EMB adaptively assigns weights to various $n$-step returns to filter high-quality learning signals. Evaluated on Atari tasks, EMB demonstrates superior sample efficiency, matching state-of-the-art baselines like Retrace($\lambda$) with 50% fewer samples. The authors conclude that this probabilistic selection mechanism offers a more robust and principled approach to long-horizon credit assignment compared to fixed-decay heuristics.

---

After reading the paper, review comments, and author responses, the AC summarizes the paper's strengths and weaknesses below.

**Strengths**
- Novel Perspectives: The paper introduces new insights into salience and stability (proposing a corresponding metric) and the bootstrapping step in multi-step RL.
- Significant Performance Gains: The proposed EMB achieves superior results on the examined Atari tasks, requiring only 50% of the samples used by the compared baselines.

**Weaknesses**
- Overclaimed Statements: The $b(Z_n)$ score and overall EM framing are viewed as hand-crafted approximations rather than a rigorous EM algorithm. Additionally, the theoretical statements only support tabular settings and lack deeper justification.
- Limited Evaluation: The experimental results are based on only three rounds and are restricted to the Atari domain. There is no evidence that the proposed method works for continuous environments.
- Outdated Baselines: The references are relatively old, with the most recent baseline from 2016. The paper provides no evidence of superiority against recent methods.
- Paper Clarity: The paper is difficult to read because of redundant paragraphs and poorly defined notations.

**Reviewer Concerns:**

Multiple reviewers shared the above concerns, and the authors provided detailed responses aiming to address them. While some issues might be resolved or alleviated, the AC still has concerns regarding the overclaimed statements of the EM design, the limited scope of evaluation tasks, and the outdated baselines.

**Reviewer Scores:**

The initial scores for this paper were [2, 4, 6, 6], indicating a diverse perspective among the reviewers. Following the author rebuttal, the reviewer with a score of 4 committed to increasing their rating to a 6, while one reviewer with a 6 expressed intent to maintain their original evaluation. The remaining reviewers provided no further response; the AC believes it is unlikely they would increase their scores even if a full discussion phase were available. In light of the remaining concerns noted above, the AC decided to reject this version of the paper. The authors are encouraged to polish the work considering these suggestions.

---

### Decision · Program_Chairs · 2026-01-26

Reject